

# Unravelling Light and Microbial Activity as Drivers of Organic Matter Transformations in Tropical Headwater Rivers

James F. Spray[1*], Thomas Wagner[1], Juliane Bischoff[1], Sara Trojahn[1], Sevda Norouzi[1], Walter Hill[1], Julian Brasche[2], Leroy James[2], Ryan Pereira[1*]

[1]The Lyell Centre, Heriot-Watt University, EH14 4AP, Edinburgh, UK.
[2]Iwokrama International Centre for Rainforest Research and Development, 77 High Street, Kingston Georgetown, Guyana

*Correspondence to*: James F. Spray (jfs8@hw.ac.uk) and Ryan Pereira (R.Pereira@hw.ac.uk)

**Abstract.** Connecting tropical rainforests to larger rivers, tropical headwaters export large quantities of carbon and nutrients as dissolved organic matter (DOM), and are thus a key component of the global carbon cycle. This DOM transport is not passive, however; sunlight and microbial activity alter DOM concentrations and compositions, affecting riverine greenhouse gas emissions and downstream ecosystems. The effects of sunlight and microbial turnover/activity on DOM concentrations and compositions in tropical headwaters are currently poorly understood, but novel Size Exclusion Chromatography (SEC) techniques coupled to suitable detectors can for the first time quantify their influences. Here, we present in-situ incubation experiments from from headwaters of the Essequibo River, in the Iwokrama Rainforest, Guyana, where sunlight oxidised up to 9% of dissolved organic carbon (DOC) over 12 hours, at higher rates than in larger tropical rivers. DOM transformations occurred in both photo-sensitive and supposedly photo-resistant pools. Microbial activity had varying, less clear influences on DOC concentrations over the same time span; compositionally, this appeared to extend beyond known bio-labile components. Biopolymers were particularly reactive to both processes. We show sunlight has the greater potential to mineralise headwater DOM and thus potentially influence degassing. Our approach provides a future template to constrain DOM transformations along river networks, identify biogeochemical activity hotspots, and improve greenhouse gas emissions estimations under changing environmental conditions.

## 1 Introduction

Global estimates of terrestrially-derived carbon to inland waters (rivers, lakes, and wetlands) range from ~5.1-5.7 petagrams of carbon per year (Pg C yr$^{-1}$) (Drake et al., 2018; Wehrli, 2013). However, only 0.9 Pg C yr$^{-1}$ is estimated to reach the ocean; ~3.9 Pg C yr$^{-1}$ is emitted to the atmosphere as carbon dioxide ($CO_2$) (Battin et al., 2008; Cole et al., 2007; Drake et al., 2018). Wetlands (in particular tropical wetlands) represent the largest non-anthropogenic source of methane ($CH_4$); they comprise 20-30% of total $CH_4$ emissions (Nisbet et al., 2014; Bousquet et al., 2011). In the context of the global carbon cycle, this $CO_2$ evasion is estimated to be on the same order of magnitude as anthropogenic fossil fuel $CO_2$ emissions ($7.9 \pm 0.5$ Pg C yr$^{-1}$), deforestation, and changes in land use ($1.0 \pm 0.7$ Pg C yr$^{-1}$) (Ward et al., 2017). However, there is significant uncertainty over its sources and controlling mechanisms (Raymond et al., 2013). One key source of this uncertainty is the extent that carbon



and nutrients from organic sources, known as organic matter (OM), flow through inland waters 'passively' (without utilization or transformation) or 'actively' (utilized and/or transformed through biotic and abiotic processes). The river continuum (Vannote et al., 1980) and flood-pulse-shunt concepts (Junk, 1989; Raymond et al., 2016) provide useful frameworks to encapsulate passive and active land-to-ocean OM transport processes. Both mechanisms likely interchange during key

moments and at hotspot locations along networks of inland waters (McClain et al., 2003), defining focal points of interest and further research. It therefore remains a challenge to constrain the rate of OM transformations at different/sliding temporal and spatial scales, and the associated release of greenhouse gases.

Rainforests are a major hotspot of the global carbon cycle; they store and release vast quantities of OM into large tropical river systems, such as the Amazon, Congo, Orinoco, or Essequibo (Battin, 1998; Pereira et al., 2014a; Richey et al., 2002; Spencer

et al., 2012). Indeed, the annual flux of dissolved organic carbon (DOC) from tropical rivers to the ocean alone accounts for ~59% of global riverine DOC flux (Huang et al., 2012). However, of the 470 Tg C yr$^{-1}$ of $CO_2$ emitted to the atmosphere by the Amazon river alone, up to 75% is estimated to derive from OM of near-channel origin, which has been mobilized to, and mineralized within, the aquatic zone (Davidson et al., 2010; Richey et al., 2002). Tropical river systems may also be significant sources of atmospheric $CH_4$ emissions (Upstill-Goddard et al., 2017). Globally, headwaters account for 70-80% of total river

networks (Gomi et al., 2002). Furthermore, first- and second-order streams comprise over 80% of the total channel length of meso-scale Amazon drainage basins (McClain and Elsenbeer, 2001). Therefore, it is necessary to better constrain the rates of biotic and abiotic transformations of OM in headwaters, where it crosses the terrestrial-aquatic interface.

Headwaters are proximal to terrestrial OM inputs, meaning their components, such as dissolved OM (DOM; which includes DOC and dissolved organic nitrogen (DON)), exhibit large and variable fluxes that directly relate to their immediate

surroundings and environmental conditions (Pereira et al., 2014a; Voss et al., 2015). The concentration and composition of DOM in such headwaters reflects both hydro-climatic changes in the relevant ecosystem and the compositional diversity of carbon pools in the surrounding soils and vegetation (Battin, 1998; Mann et al., 2014; Seidel et al., 2015; Spencer et al., 2016). These factors influence the biotic and abiotic lability or recalcitrance of DOM components entering the rivers, and will likely control their fate, through degassing, transport, and burial (Amon, 2002).

Two of the main processes that influence, and are influenced by, the concentration and composition of OM in rivers are photodegradation and microbial activity/degradation (biodegradation) (Cory and Kling, 2018; Jones et al., 2016; Mostofa et al., 2013a; Obernosterer and Benner, 2004; Wiegner and Seitzinger, 2001). Photodegradation refers to the process whereby ultraviolet (UV) and visible light breaks down OM. Some DOM compounds contain chromophores, defined as functional groups in organic compounds that can absorb photons with an efficiency sufficient to promote an electron from a ground to an

excited state (Mostofa et al., 2013a). Chromophoric structures that are comprised of conjugated double bonds can be broken down upon absorbing this radiation, leading to the photodegradation of the molecule in question (Jones et al., 2016; Zepp, 1988). This process is thought to mainly effect Coloured DOM (CDOM). However, there is debate over the extent that photodegradation results in the complete mineralization of DOM to $CO_2$. It may only achieve partial degradation, breaking down coloured, high molecular weight (HMW) compounds into low molecular weight (LMW) components that are potentially



more labile (Cory et al., 2014; Jones et al., 2016; Lou and Xie, 2006; Mostofa et al., 2013b; Obernosterer and Benner, 2004; Remington et al., 2011). In addition to $CO_2$ outgassing, the nitrogenous photoproducts ammonium ($NH_4^+$), nitrate ($NO_3^-$), and nitrite ($NO_2^-$) are also released by the photodegradation of humic substances and DON (Mostofa et al., 2013a); these compounds are critical for biological productivity (Zepp, 2003). Microbiological cycling of OM represents an alternative/competing process that also alters the abundance and composition of DOM (Mostofa et al., 2013b). The process of

biodegradation comprises the consumption of DOM compounds by microbes, leading to DOM mineralization and $CO_2$ production via respiration (Cory et al., 2014). LMW aliphatic DOM is more easily broken down by microorganisms than larger, more aromatic and complex molecules (Jones et al., 2016).

It has been posited that these two processes can combine through the mechanism of photo-simulated biodegradation. Here, larger, more recalcitrant CDOM compounds are partially photodegraded into smaller, more labile compounds. This facilitates

further biodegradation, as the latter are more easily consumed by microorganisms (Cory et al., 2014; Moody and Worrall, 2016; Mostofa et al., 2013a).

The quantification and characterization of OM degradation is often limited through commonly used analytical technologies, which focus on CDOM. The UV-visible absorbance of river water is often used as a proxy for DOC concentrations on global scales, based on the relationship between CDOM and DOC (Adams et al., 2018; Carter et al., 2012; Griffin et al., 2011; Helms

et al., 2008; Weishaar et al., 2003). This method ignores variations in the concentration of optically invisible DOM (iDOM), however. Fourier-transform ion cyclotron resonance mass spectroscopy (FT-ICR-MS), meanwhile, can reveal compositional variations in DOM, but is difficult to align with quantitative DOC changes, and excludes biopolymers, a key DOM component. Novel Size Exclusion Chromatography (SEC) techniques coupled to suitable detectors present a unique alternative to quantitatively study compositional DOM transformations (Huber et al., 2011), as they can determine the concentrations of

different DOM components. Recently, SEC has been used to reveal that iDOM comprises up to 89% of the DOM fraction in tropical headwaters and varies greatly over short timescales (Pereira et al., 2014a), for example; rapid changes in DOC composition and flux have also been reported for a large Arctic river (Voss et al., 2015).

Given the importance and apparent compositional variability of DOM in tropical headwaters, it is important to understand the processes that influence DOM concentration and composition. These processes could have significant implications both

upstream and downstream, for factors including carbon fluxes, bioreactivity, further $CO_2$ evasion, and oxygen availability (Soares et al., 2019). Though several studies have examined the effect of photodegradation on riverine DOC concentrations (Amon and Benner, 1996; Moody and Worrall, 2016; Mostofa et al., 2005a; Pickard et al., 2017; Pullin et al., 2004; Spencer et al., 2009; Ward et al., 2013), few have examined compositional changes of DOM (Voss et al., 2015), and to our knowledge none have quantified the effects of photodegradation on both compositional DOM changes and DOC concentrations in tropical

headwaters.

One approach to ascertain the importance of photodegradation and biodegradation of DOM in tropical headwaters is to isolate and incubate water, and establish how the DOC concentration and DOM composition changes over time, and by what dominant process (photochemistry versus microbial activity). Here this approach was adopted on water samples collected from three



sites in the headwaters of the Essequibo River, within the Iwokrama Rainforest, Guyana (Fig. 1). Splitting the initial sample
and only exposing half to sunlight allows us to isolate and quantify photochemical and microbial influences (Fig. 2a).
Furthermore, by analysing the resulting samples using SEC, for the first time we can quantify not only changes in DOC
concentrations, but also degradation-influenced compositional changes in DOM.

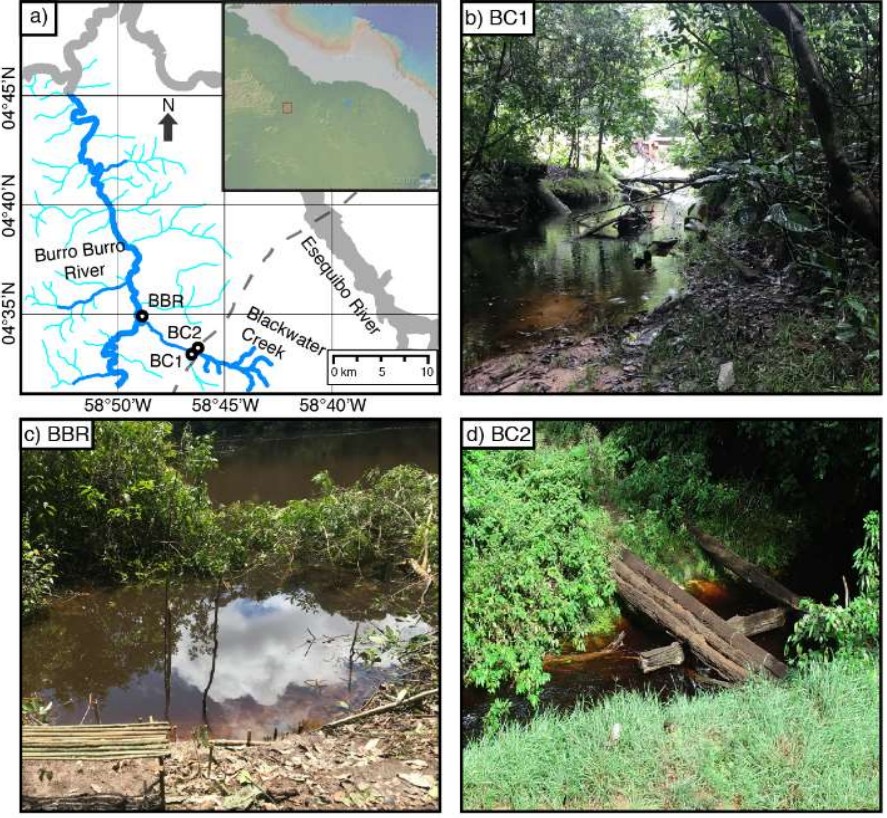

**Figure 1. (a) Location of sample sites within the Iwokrama Rainforest, Guyana (modified from Pereira et al., 2014; inset created**
**using http://www.geomapapp.org). (b) Blackwater Creek, Site 1 (BC1; headwater stream, shaded conditions); (c) Burro Burro River**
**Site (BBR; open conditions); and (d) Blackwater Creek, Site 2 (BC2; headwater stream, open conditions).**



## 2 Methods

### 2.1 Sampling strategy & procedure

In-situ irradiation experiments were conducted at the Iwokrama tropical rainforest in central Guyana, South America (04° 40'
N, 58° 45' W), based from the Iwokrama International Centre for Rainforest Conservation and Development (IIC). Three
different study sites were identified, one located on the Burro Burro River (BBR), a tributary of the Essequibo connecting the
interior with the tropical Atlantic coastline, and two on the Blackwater Creek (BC1 & BC2), which is a headwater that feeds
into the BBR (see Fig. 1). The DOM compositions of both sites have previously been explored (Pereira et al., 2014a). The

experiments were performed in situ, i.e. within the water column at each location, rather than in a laboratory under controlled
light and temperature conditions, for two reasons. First, this approach allows to study real-world conditions – including a full
light spectrum, dappled light, changes in sunlight angle, and temperature fluctuations. Second, this approach is particularly
useful in remote regions such as the Essequibo River in central Guyana, where controlled irradiation experiments using ex situ
approaches are logistically impractical. By their nature, headwaters in the Amazon system will be in remote areas with little

infrastructure, and so the robust, portable, and simplified nature of our approach allows potential for comparison experiments
to be run elsewhere.

   At each site, experiments were conducted to determine how photochemical and microbial / 'light' and 'dark' processes affect
the DOC concentration and composition of river water over time. In each case, homogenized river water was collected in pre-
cleaned 20 L carboys rinsed with copious amounts of sample water at 7 am (sunrise), and 180 ml was subsampled in two 60

ml HDPE bottles to provide an initial ($t_0$) sample. The remaining water was then used to rinse and fill two opaque polyethylene
containers (6 L in each container), which were modified with floatation devices that permitted free floatation within the water
column at each site, while anchored to a set location. One container was covered with an opaque lid ('dark') to act as a dark
control, whereas the other was left exposed to sunlight ('ambient'); the 'dark' container had ventilation holes to allow for free
atmospheric exchange (see Fig. 2a).

Further 180 ml subsamples were taken from each container after 12 hours, by which time the sun had set. Following this, a lid
was placed onto the 'ambient' container and both containers were left overnight. A further subsample was then taken from
each container the next morning (after 22 hours in total) to address the debate over whether photo-simulated biodegradation is
an important influence in headwaters (Cory et al., 2014; Moody and Worrall, 2016). We consider that photo-stimulated
biodegradation should occur overnight in the 'ambient' container, but not the 'dark' container, as partial photodegradation

should only have occurred in the former. This process would be evidenced by further decreases in DOC concentration in the
'ambient' container overnight, relative to the 'dark' container. Running the incubation overnight also allows us to calculate
changes in DOC concentration and DOM composition over an approximated day-night cycle.

   This incubation procedure was conducted three times at each site (denoted a, b, and c, e.g. BC1a, BC1b, and BC1c) with
incubations being run on successive days (See Table S1 for starting dates of each incubation). The volume of water remaining

in each container was measured after the completion of each experiment to allow subsequent DOC concentrations to be



corrected for evaporation. In the event of rainfall, the lid was temporarily placed upon the 'ambient' container until it had stopped, and the timing of placing and removing the lid was recorded (time lost to rainfall ranged from 0-4 hrs). To allow comparison, the changes seen in each ambient experiment were corrected for this lost time.

Each subsample was measured for temperature, pH, conductivity, total dissolved solids (TDS), salinity, and conductivity, using
a Hach® Pocket Pro+ Multi 2 Tester, and for oxidation reduction potential (ORP) using a Hach® Pocket Pro+ ORP Tester, which were calibrated daily. Each subsample was then filtered through a pre-combusted (450 °C for 8 hours) GF/F (0.7 µm) filter. The filtered water was decanted into two 60 ml HDPE bottles, which were then kept in dark conditions to prevent any further photodegradation. One bottle was kept cool at 4 °C for analysis at a mobile laboratory setup at the IIC and another bottle was frozen in preparation for transport back to the Lyell Centre, UK, for SEC analysis, following storage protocols by
Heinz and Zak (2018). Over the course of each experiment, UV and visible irradiance at each sample site was measured at hourly intervals using SpectriLight® ILT950 and ILT950UV Spectroradiometers, coupled together to produce a single spectrum from 200 to 700 nm.

**2.2 Analysis**

The DOC concentration of each sample was measured at the IIC laboratory using a Sievers M5310C Portable TOC Analyser, with attached GE Autosampler; this instrument has a detection range of 0.03 ppb to 50 mg $L^{-1}$. The instrument was calibrated prior to field deployment following the manufacturers specification, and pre-weighed sucrose check standards (Sigma Aldrich, Supelco #47289) were freshly dissolved with 18.2 M Ohm deionised carbon-free water on site at the IIC and analysed periodically throughout the field campaign to ensure instrument accuracy and precision; these standards demonstrated
reproducibility better than 3% RSD. Measurements on each water sample were performed in triplicate (from which SD was obtained), and samples were blank corrected with 18.2 M Ohm deionised carbon-free water (one blank was analysed for every sample run, with ~20 samples per run). Concentrations from the 'ambient' container were then corrected for evaporation. UV-Vis Absorbance was measured using a Biochrom® WPA Lightwave II UV/Visible Spectrophotometer. Measurements were performed in triplicate, within 24 hrs of sampling in each case. DOM composition was determined at the UK Lyell Centre
laboratories on thawed -20 °C samples using a novel LC-OCD-OND (Liquid chromatography with organic carbon detection and organic nitrogen detection) that allows ~1 ml of whole water to be injected onto a size exclusion column (SEC; 2 ml $min^{-1}$; HW50S, Tosoh, Japan) with a phosphate buffer (potassium dihydrogen phosphate 1.2 g $L^{-1}$ plus 2 g $L^{-1}$ di-sodium hydrogen phosphate x 2 $H_2O$, pH 6.58) and separated into five "compound-group specific" DOM fractions. The resulting compound groups are identified using unique detectors for OC, UV-amenable carbon and ON (Huber, et al. 2011). All peaks were
identified and quantified with bespoke software (Labview, 2013) normalized to International Humic Substances Society humic and fulvic acid standards, potassium hydrogenphthalate and potassium nitrate. The LC-OCD-OND is also able to quantify organically bound N in the humic substances (HS) and biopolymer compound groups, in addition to inorganically bound $NH_4^+$ and $NO_2^-$, by passing the sample through a UV-reactor that converts them to $NO_3^-$ (Huber et al., 2011).



### 2.3 Calculations


To calculate the photochemical influence on net DOC concentration, the values from the 'ambient' container, which theoretically represent the combined effects of both photochemical and microbial influences, were corrected by subtracting the values from the 'dark' container (microbial influences only; see Fig. 1e).

The LC-OCD-OND data were used to calculate the relative proportions of the five DOC component groups outlined above:

biopolymers, HS, building blocks, LMW neutrals, and LMW acids. The DOC concentrations were then used to calculate absolute concentrations for each component group in the 'ambient' and 'dark' samples, which were then subtracted as described above to obtain the apparent photochemical effect.

The LC-OCD-OND data can also be used to further interrogate the properties of the HS compound group. Previous studies have identified a correlation between the aromaticity (specific absorption coefficient (SAC)/OC, as obtained by the UV-

detector of the LC-OCD-OND) and molecularity (molecular weight) of the HS fraction (Huber et al., 2011; Huber and Frimmel, 1996). Plotted against each other in a HS-diagram, as introduced by Huber and Frimmel (1996), this relationship can help distinguish between aquagenic and pedogenic origins of HS. Hence, HS aromaticity and HS molecularity were calculated from the LC-OCD-OND data following the methods of Huber et al. (2011).






# 3 Results

## 3.1 Photochemical and microbial degradation of DOC

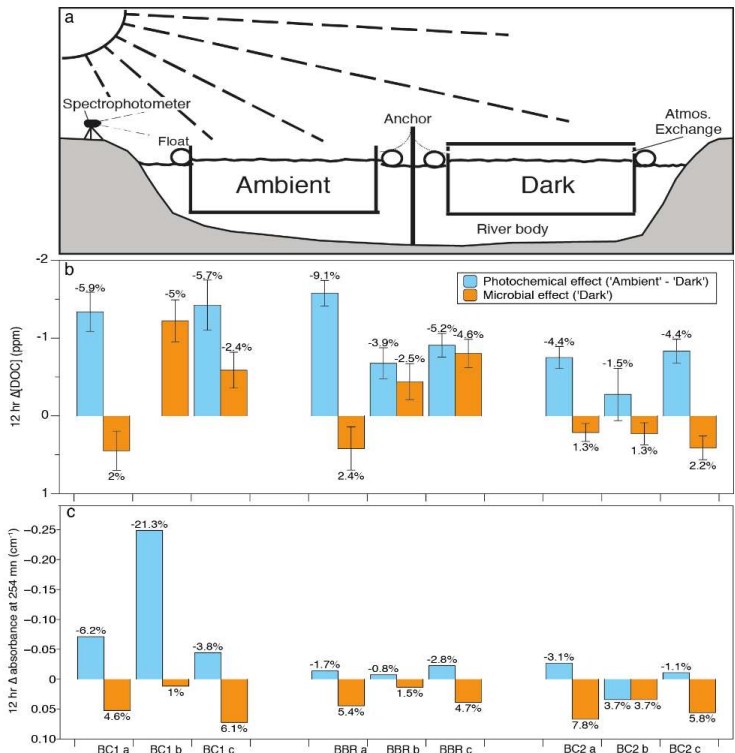

**Figure 2. (a)** Experimental setup. **(b)** Changes in DOC concentration across the 12 hr study period during the incubation experiments
at Sites BC1, BBR, and BC2 (suffixes a, b, and c denote repeat number of each experiment at each site; note photochemical data for
BC1b were lost due to measurement error). Error bars show standard deviation. Change as a percentage of initial DOC
concentration is also shown. **(c)** Changes in absorbance at 254 nm across the 12 hr study period. Note inverted y axes.

Photochemical mineralization of DOC appears to have occurred in each of the incubations across all three sites, with the
exception of BC2-b (Fig. 2b, blue bars), which exhibited no change exceeding sample SD. Excluding this incubation,
measureable (i.e., > 0.3 ppb) and significant (i.e., exceeding both sample SD and 3% RSD from repeated analysis of standard)
decreases of 0.7-1.6 ppm (~4-9% of initial DOC concentrations) were observed over the 12-hour interval in each of the other
incubations. The upper limit of this degradation exceeds those observed in previous studies on river water from the main stems



of the Rio Negro, which reported photodegradation of ~5% after ten hours of exposure to sunlight (Amon and Benner, 1996),
and the Congo River, which showed ~10% photodegradation over a two-day period (Spencer et al., 2009).

The observed photodegradation of DOC was accompanied by decreases in absorbance at 254 nm (a proxy for CDOM) in each incubation (Fig. 2c, blue bars), with the exception of BC2-b (which did not show any change in DOC within error). These decreases in DOC and CDOM were positively correlated ($R^2$=0.78, p<.05). Interestingly, however, the specific UV absorbance at 254 nm ($SUVA_{254}$) of all but one incubations increased due to photochemical influences (~2-5%, excluding BC1-a; Table
1). It appears then that though photobleaching occurred, non-aromatic compounds were preferentially removed by sunlight over aromatic DOC.

The microbial effect on DOC concentration during the incubations was more varied and less clear than that of photochemistry (Fig. 2b, orange bars), with biodegradation appearing to occurr in four incubations, and DOC production being apparent in five of the incubations. All changes were measurable and exceeded sample SD, but only the decreases observed in incubations
BC1-b and BBR-c were large enough to exceed the precision as determined by repeated analysis of standards (3% RSD). Any additional DOC likely either came from (potentially microbial or abiotic) breakdown of POC, or conversion of DIC to DOC (e.g. primary production). Absorbance at 254 nm (Fig. 2c, orange bars) increased in every incubation. Taken in isolation, this might suggest that any DOC being produced through microbial action was CDOM, and/or that iDOM was selectively utilized by biota over CDOM (Amado et al., 2007; Moran et al., 2000). The accompanying increases in $SUVA_{254}$ during each
incubation (~2-10%; Table 1) likewise suggest the preferential breakdown of non-aromatic substances and/or the production of aromatic compounds; however, as shown below, this situation is more complex.

**Table 1. pH and $SUVA_{254}$ of initial, 'Ambient' (A), and 'Dark' (D) samples, cumulative irradiation, initial DOC concentration, initial turbidity, and initial absorbance at 254 nm. Error shown is SD.**

| Incubation | pH | | | $t_0$ Turbidity (NTU) | Cum. Irradiation (MJ m$^{-2}$) | $t_0$ [DOC] (ppm) | $t_0$ absorbance at 254 nm (cm$^{-1}$) | $SUVA_{254}$ (L mg$^{-1}$ m$^{-1}$) | | |
|---|---|---|---|---|---|---|---|---|---|---|
| | $t_0$ | A | D | | | | | $t_0$ | A | D |
| BC1-a | 4.88 | 4.84 | 4.86 | 0.83 ± 0.04 | 0.36 | 22.4 ± 0.2 | 1.14 | 5.10 | 5.22 | 5.23 |
| BC1-b | 4.48 | 4.59 | 4.53 | 0.84 ± 0.03 | 0.30 | 24.6 ± 0.2 | 1.17 | 4.75 | - | 5.05 |
| BC1-c | 4.45 | 4.43 | 4.52 | 0.48 ± 0.01 | 0.25 | 24.9 ± 0.2 | 1.18 | 4.74 | 5.27 | 5.15 |
| BBR-a | 5.65 | 5.88 | 6.02 | 2.35 ± 0.02 | 10.99 | 17.3 ± 0.2 | 0.83 | 4.79 | 5.17 | 4.93 |
| BBR-b | 5.54 | 5.73 | 5.62 | 2.79 ± 0.04 | 41.97 | 17.5 ± 0.2 | 0.87 | 4.98 | 5.35 | 5.19 |
| BBR-c | 5.51 | - | - | 2.77 ± 0.01 | 51.08 | 17.5 ± 0.2 | 0.83 | 4.73 | 5.34 | 5.19 |
| BC2-a | 4.56 | 4.44 | 4.34 | 0.55 ± 0.02 | 20.70 | 17.0 ± 0.1 | 0.86 | 5.07 | 5.48 | 5.39 |
| BC2-b | 4.62 | 4.24 | 4.24 | 3.68 ± 0.09 | 8.93 | 18.0 ± 0.1 | 0.92 | 5.11 | 5.48 | 5.23 |
| BC2-c | 5.52 | 4.37 | 4.37 | 0.63 ± 0.01 | 7.22 | 19.0 ± 0.1 | 0.96 | 5.07 | 5.40 | 5.25 |

Photochemically-induced DOC decreases were broadly similar within and between each site (Fig. 2b), demonstrating that the rate of photodegradation appears to be comparable between smaller upper headwaters (BC) and larger downstream tributaries



(BBR), despite their hydrological differences (BC is a second-order headwater with a length of ~6 km and a catchment of ~20 km$^2$; BBR is a fifth-order tributary with a length of 90 km and a catchment of ~3,200 km$^2$) (Pereira et al., 2014b). The total cumulative irradiation at the shaded site BC1 (0.25-0.35 MJ m$^{-2}$) was approximately two orders of magnitude lower than those

of the more exposed sites BBR and BC2 (~10-50 MJ m$^{-2}$) (Fig, 1; Table 1). For comparison, six hours of sunlight recorded in a previous study in the Amazon (Amado et al., 2006) equated to ~1 M Jm$^{-2}$. The similar degradation observed between sites, despite disparity in the cumulative irradiation, suggests that cumulative irradiation alone did not control the degree of photodegradation at the three sites. Turbidity and pH may also moderate photodegradation. Turbidity can limit the attenuation of sunlight into the water column (Helbling and Zagarese, 2007), and thereby photodegradation. Though the turbidity values

of incubations BBR-a & c were higher than that of BC1-a & c, those of BC2-a & c were lower than those of BC1, implying that low turbidity is unlikely to have caused the relatively high yield of photodegradation at BC1 (Table 1). Low pH can enhance the breakdown of DOC as acidic conditions are more favourable for Fenton's reactions, a pathway for photodegradation (Molot et al., 2005; Wu et al., 2005). Conversely, DOC may also be driving the low pH at each site to some extent – OM has been shown to buffer pH in acidic conditions in a boreal headwater (Köhler et al., 2002). However, incubations

from BC1 and BC2 had similar pH values, but different photodegradation when normalized for irradiation (Table 1). It appears therefore that the amount of irradiation received or initial physical conditions of river water cannot satisfactorily explain the degree of photodegradation measured in each incubation.

The concentration and nature of DOM has also been suggested to affect the effectiveness of photochemical and microbial degradation; elevated DOC concentrations have been shown to increase photodegradation (Jones et al., 2016). This is

consistent with our findings; the initial DOC concentrations at Site BC1 were ~3-5 ppm higher than at Sites BBR and BC2 (Table 1). From a compositional viewpoint, high CDOM concentrations in aquatic environments may reduce the rate of photodegradation, as chromophoric structures can have a photo-shielding effect, thereby limiting the attenuation of light (Artifon et al., 2019; Mostofa et al., 2013a). Increased CDOM concentrations have also been found to be positively correlated with higher DOC photodegradation, however, as chromophoric structures can be broken down by solar radiation (Jones et al.,

2016). This is also the case in our studies – the initial absorbance at 254 nm for incubations at site BC1 was higher than those conducted at sites BBR and BC2. However, the photochemical breakdown of CDOM may not always lead to the complete mineralization of DOC. To elucidate the influences of photochemical and microbial degradation, we next explore compositional changes in DOM.

**3.2 Photochemical and microbial driven changes in DOM composition**

SEC, coupled with suitable detectors, permits the quantification of changes in the composition of DOM. Specifically, five compound groups can be identified based on their molecular weight and optical properties: (i) biopolymers (likely hydrophobic, high molecular weight >> 20.000 g mol$^{-1}$, largely non-UV absorbing extracellular polymers); (ii) HS (higher molecular weight ~ 1000 g mol$_{-1}$, UV absorbing); (iii) building blocks (lower molecular weight 300-500 g mol$^{-1}$, UV absorbing



humics); (iv) LMW 'neutrals' (350 g mol$^{-1}$, hydro- or amphi-philic, non-UV absorbing); and (v) LMW acids (350 g mol$^{-1}$).
Determining the composition of DOM is important as it can be indicative of both its bioavailability (bio-lability), and its
susceptibility to photodegradation (photo-lability). HMW, chromophoric compounds with relatively high aromaticity are
thought to be biologically recalcitrant but photo-labile (Artifon et al., 2019; Mostofa et al., 2013a; Sharpless et al., 2014),
whereas LMW, optically-invisible molecules with lower aromaticity are bio-labile but may be less susceptible to

photodegradation (Koehler et al., 2012; Stubbins et al., 2010). Some of these bio-labile molecules may themselves be photo-
products, created by incomplete photodegradation of larger photo-labile DOM (Stubbins et al., 2010).

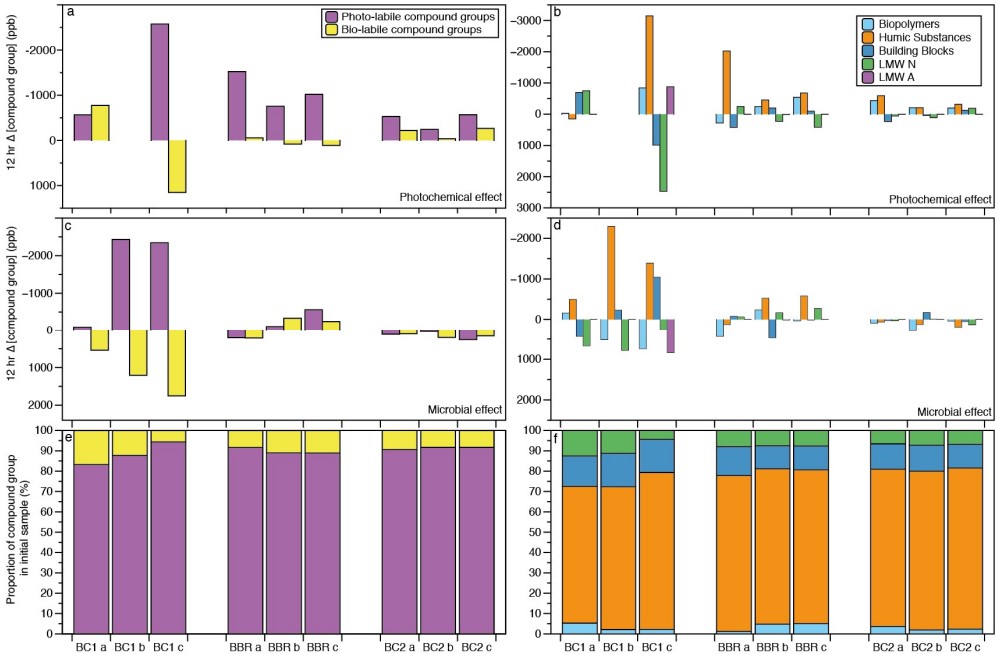

**Figure 3.** a) Photochemistry-induced, and b) microbial-induced changes in the concentrations of photo-labile (HS, protein-based
biopolymers and building blocks) and bio-labile compounds (non-protein-based biopolymers, LMW neutrals, and LMW acids) over
12hr incubation period; c) initial proportions of photo-labile and bio-labile compounds. D) Photochemistry-induced, and e)
microbial-induced changes in the concentration of each DOM component over 12 hrs of incubation; f) DOM composition of initial
samples for each incubation. Note inverted y axes.

Figure 3a shows the changes in photo- and bio-labile DOM compound groups over 12 hours, as a result of photodegradation.
Photo-labile compound groups (which comprised 80-90% of the initial samples; Fig. 3e) were removed in every incubation,
whereas bio-labile compound groups were removed in some incubations but added in others. This suggests that LMW,
optically invisible DOM can be photodegraded (e.g. BC1-a). Furthermore, photodegradation appears to be capable of both
complete mineralization of photo-labile compound groups, and their partial degradation into photo-products (e.g. BC1-c). The
variation seen in bio-labile compound groups (Fig. 3a) may therefore represent a balance between their mineralization and



addition (through partial photodegradation of photo-labile components). As a percentage of initial concentrations, the

variations in the bio-labile pool were of similar or greater magnitude than in the photo-labile pool (Fig. S2b), implying that the latter is not necessarily more reactive to photodegradation. Therefore, here we show that photodegradation does indeed break down photo-labile DOM. However, complete mineralization, combined with sunlight's ability to also remove supposedly photo-resistant DOM (Stubbins et al., 2010), means that photodegradation in tropical headwaters does not necessarily improve the bioavailability of DOM transported downstream. It does, however, decrease the total amount of DOM being exported.

These compositional changes can be explored in greater detail – Fig. 3b shows that in all experiments bar BC1-a, the breakdown of photo-labile DOM was largely driven by HS (which was the most abundant compound group in initial samples; Fig. 3f). Likely HS breakdown products would be building blocks and LMW neutrals. The former; however, is comprised of UV-absorbing compounds, and so would also be expected to break down upon exposure to UV/visible irradiation. This may explain why its concentration increased in incubations where large quantities of HS were degraded (e.g., BC1-c, BBR-a),

whereas BC1-a, where there was a (relatively small) increase in HS, showed the largest decrease in the concentration of building blocks. It is interesting to note that the initial proportion of HS in incubation BC-1a (67%) was lower than that of all other runs (76-79%; Fig. 3f). The initial composition of HS for this incubation may have been relatively resistant to photodegradation. The photochemical and/or microbial breakdown of POC can recharge DOC, which could also have recharged the HS compound group (Mayer et al., 2006; Mostofa et al., 2013b; Porcal et al., 2015).

Figure 3c shows that the bio-labile and photo-labile (theoretically bio-recalcitrant) DOM compound groups showed varied responses to microbial action, mirroring the varied response in DOC concentration (Fig. 2b). Despite its relative availability for microbial respiration, bio-labile DOM was only removed from two incubations (BBR-b, c, of which only BBR-c showed a change in DOC exceeding instrument RSD). The availability of LMW, bio-labile compound groups in initial samples (Fig. 3f) did not appear to influence the degree of microbial degradation. Further, HMW, supposedly recalcitrant DOM was

apparently susceptible to microbial degradation – indeed, for the two incubations that demonstrated the highest net microbial degradation (~4-5%; BC1-c and BBR-c; Fig. 2), more photo-labile DOM was removed than bio-labile DOM (Fig. 3c). Therefore, the observed increases in absorbance at 254 nm do not necessarily reflect the preferential removal of non-UV absorbing DOM. While over longer timescales, previous studies have shown that microbes can process HMW, photo-labile DOM where bio-labile DOM is limited; this also appears to be the case here (Koehler et al., 2012).

Figure 3d shows that, within the photo-labile pool, HS was removed by microbial activity in five of the incubations, including BC1-c and BBR-c, at a similar magnitude to that of photodegradation. It appears therefore, that even HMW molecules within this compound group are bio-available, not just smaller molecules (building blocks). LMW acids were only observed to any significant extent in Experiment BC1-c (Fig. 3d); they were formed through microbial processes, as none were present in the initial sample (Fig. 3f).

These findings have significant implications for understanding rivers as biogeochemical reactors- on short timescales. Microbial activity does not appear to be dictated by the lability of the DOM pool in tropical headwaters, but instead is capable



of altering a variety of DOM compositional pools on daily timescales, through both biodegradation and bio-accumulation/primary production.

Though biopolymers represented a relatively minor proportion of initial samples (~1-6%; Fig. 3f), relative to initial
concentrations, this compound group was highly reactive to both photochemical and microbial influences (Fig. S2a, c). This compound group contains both UV and non-UV containing compounds, which may explain this relatively high reactivity under both stimuli. Though detectable through SEC, biopolymers are not measured by FT-ICR-MS, further highlighting the importance of the former. The biopolymer compound group likely contains proteins and amino sugars, which are important for biota, and polysaccharides, which are a key component in microbial biofilms (Huber et al., 2011; Her et al., 2002; Flemming
et al., 2007). The apparent high reactivity of this compound group may therefore reflect complex microbial processes, and warrants further investigation.

The above conclusions, combined with the revelation that photodegradation does not always yield bio-degradable products, would suggest that photo-simulated biodegradation (Cory et al., 2014) is unlikely to be an important process in tropical headwaters on short (daily) timescales in upper headwaters. This was shown to be the case here; there was no consistent change
in the DOC concentrations of the 'ambient' incubations overnight, following their exposure to sunlight: five of the incubations showed no change overnight exceeding SD, two incubations (BC1-c and BBR-c) experienced a slight increase in DOC (0.5-0.6 ppm), and only one incubation (BBR-a) showed further decreases in DOC (-0.5 ppm) that would be expected had photo-stimulated bacterial degradation occurred overnight (Fig. S1). A previous study also found no evidence of photo-stimulated bacterial degradation over a similar timescale in a temperate peatland headwater (Moody and Worrall, 2016).


### 3.2.1 Revisiting the stability of HS in headwater rivers

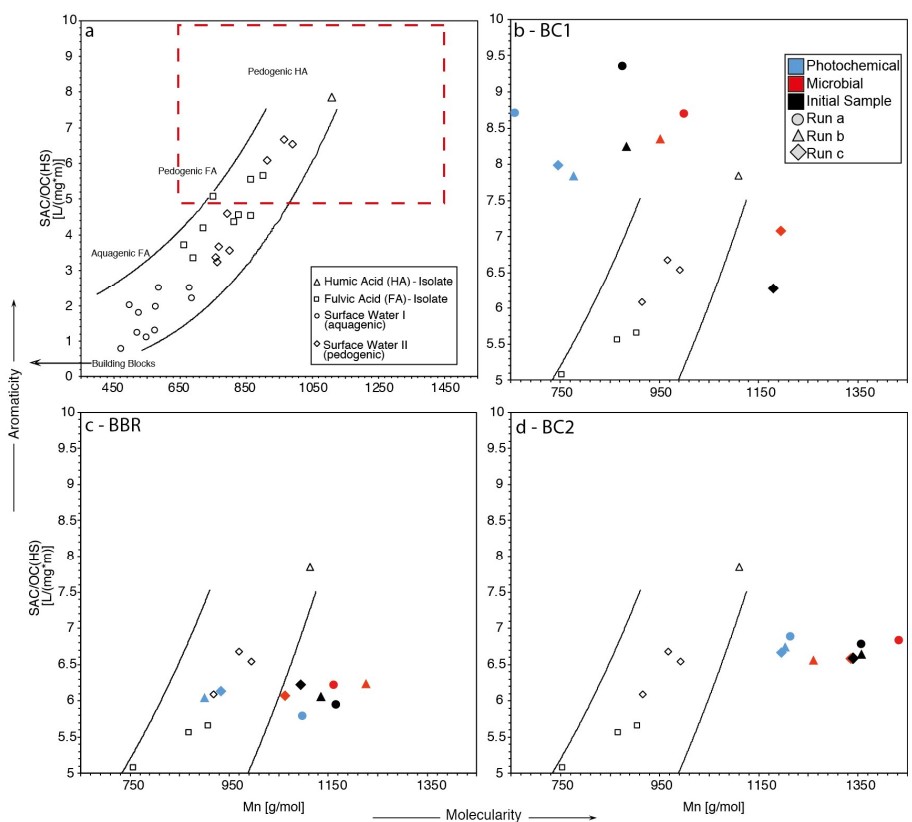

**Figure 4. HS diagrams showing aromaticity and molecularity of the HS fraction for the initial samples from each incubation and samples after 12 hrs of exposure to photochemical and microbial influence (b-d). (a) Also plotted are data for standards and**
**representative lake and river samples from Huber et al. (2011). Dashed red box in (a) shows region of plots (b-d).**

Figure 3f shows that HS was the most common compound group in riverine DOM at each site. Previous studies have explored compositional changes and inherent stability of HS by examining the relationship between aromaticity – i.e. the specific absorption coefficient (SAC) normalized to OC – and molecularity (molecular weight) (Huber et al., 2011). This approach
shows that all samples exhibited high aromaticity and molecularity, indicative of a pedogenic origin (Fig. 4). Examining Fig. 4 for every incubation, the molecularity of the HS compound group decreased following photodegradation, compared to the initial sample. This confirms that sunlight breaks down larger organic molecules into smaller components, both within the HS compound group and between DOM compound groups (Mostofa et al., 2013a). As anticipated, we observe no such clear change in molecularity in response to microbial action (Fig. 4, red points), in line with the more varied responses of DOM



composition and DOC concentration to microbial action (Figs. 2b, 3d). While there was no consistent trend, changes in molecular weight during microbial incubation were similar in magnitude to those observed under photodegradation, suggesting microbes are capable of both the breakdown and self-assembly of DOC within the HS component group on daily timescales (Xu and Guo, 2018).

Unlike for molecularity, there was no consistent decrease in aromaticity accompanying photodegradation (Fig. 4), nor was

there a consistent change in aromaticity in response to microbial activity across the incubations. The changes in UV absorbance at 254 nm seen in Fig. 2c may therefore have been driven by aromaticity changes outside of the HS compound group. Photochemically-driven changes in aromaticity did not correspond with observed changes in DOC (Fig. 2), suggesting that the aromaticity of the HS compound group is relatively unsusceptible to photodegradation. Further examination of the interaction between aromatic compounds and sunlight within the HS compound group is needed, as previous studies have reported a loss

of DOM aromaticity in response to sunlight (Sharpless et al., 2014).

The initial sample of BC1-a, which was the only incubation not to experience photodegradation of HS (Fig. 3b), had the highest aromaticity and lowest molecularity of any of the samples analysed (Fig. 4b). We can only speculate, but it seems possible that the lower molecularity, combined with the lower overall proportion of HS in the initial sample (67%, vs. 76-79% for other samples; Fig. 3f), implies that the HS compound group for BC1a was more resistant to degradation than for the other samples;

possibly it had undergone more photodegradation prior to incubation. If the more labile compounds of the HS compound group in this incubation had already broken down into building blocks, this could explain why photodegradation in BC1-a was largely achieved through degradation of building blocks, not HS (Fig. 3b).





**3.3 Organic Nitrogen Dynamics of DOM**

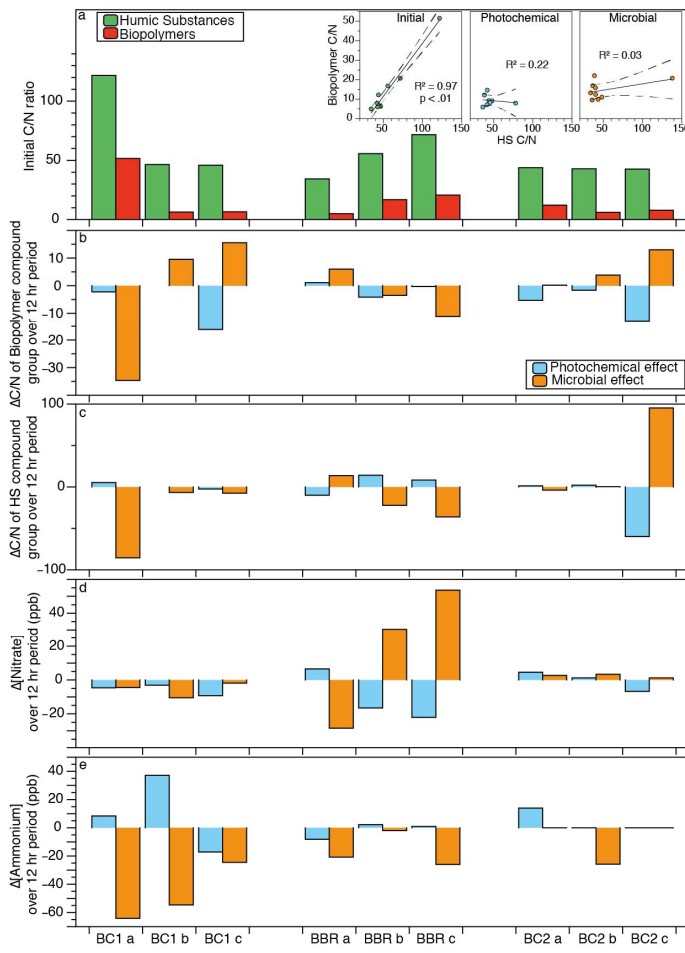


**Figure 5. (a) Initial C/N Ratios for HS and biopolymers (inset shows relationships between components before and after each incubation); Photochemistry- and microbial-driven changes in C/N ratios in (b) the biopolymer fraction and (c) the HS fraction over 12 hr incubation. Photochemistry- and microbial-driven changes in (d) nitrate and (e) ammonium during incubation.**

Addressing changes in DON may help to explain the response of DOM to microbial and photochemical influences. In addition to influencing net DOC concentration and DOM composition, photodegradation can also break down DON, leading to the formation of Dissolved Inorganic Nitrogen (DIN) in the form of nitrate and ammonium, which are critical for biological productivity (Zepp, 2003). Furthermore, the availability of DIN has been shown to limit the biological metabolism of DOM, whereas DON has been shown to be utilized by bacteria at a higher rate than DOC, affecting the rate at which they are





biologically processed (Wiegner and Seitzinger, 2001) and thereby altering the C/N ratio. Therefore, we tested if
       photochemistry or microbial activity altered C/N ratios in the biopolymer and HS component groups, and whether this resulted
       in an increase in the concentrations of nitrate and ammonium. We also explore whether initial concentrations of DON and DIN
       can inform microbial activity.

       The initial availability of DIN could not explain the variation in the microbial-driven changes in DOC (Fig. 2a, Table S4), nor
could initial DON concentrations in biopolymers and HS explain the observed changes in concentration for these component
       groups. In the initial samples from each incubation, the C/N ratios of the HS and biopolymer component groups were highly
       correlated ($R^2$ =0.97; Fig. 5a); however, this correlation disappeared following incubation. The influence of terrestrial inputs
       in tropical headwaters may maintain the partitioning of DON in HS and biopolymers, but once disconnected from this input
       the C/N ratios may diverge via a complex combination of factors that cannot be disentangled via the current approach. For
both biopolymers and HS (Figs. 5b, c), there was no clear trend in C/N ratios during each incubation, for either photochemical
       or microbial influences. This suggests that DON in these fractions responds differently than non-N containing compounds, but
       not in a consistent way across each incubation.

       Changes in the C/N ratios from neither fraction correlated with observed changes in ammonium or nitrate (Figs. 5d, e). We
       therefore conclude that the concentrations of these components were driven by other processes, such as breakdown of
particulate sources, aquatic biomass, or DON from other DOM components (e.g., building blocks, LMW neutrals, or acids)
       (Kellerman et al., 2015). Furthermore, we determined that aliphatic DOM, rather than aromatic DOM, was more commonly
       associated with N. The lack of a clear trend in microbial-driven changes in nitrate (Fig. 5d, orange bars) highlights the complex
       nature of N dynamics presented in this study. Nitrate and ammonium decreased in concentration across many of the incubation
       experiments; we envision that (photochemistry-induced) microbial scavenging may have outstripped their production via the
photochemical breakdown of DON (Zepp, 2003); this concept could be tested in future by filtering water prior to incubation
       to remove bacteria. Though evidently this study was able to capture significant degradation of DOC on daily timescales,
       photochemically and/or microbial-driven N cycling between organic and inorganic pools may be occurring at sub-daily
       timescales, i.e., too rapid to be characterized by our approach (Kellerman et al., 2015).

**4 Discussion**

       In our approach above we investigated effects of photochemical and microbial influences on riverine DOC separately;
       however, our analysis also allows for the investigation of their combined effects in 'real world' conditions over a day-night
       cycle. There was a similar extent of combined degradation in Blackwater Creek (~1-7%), a headwater stream, and the Burro-
       Burro River, a larger tributary (~1-7%; Fig. 6a); our results as presented above show that photodegradation was largely
responsible for these decreases in DOC (Fig. 2). This similarity suggests that we may be able to model a relatively constant
       daily degradation rate of DOC for these scales of river. This approach would be aided by the fact that headwaters comprise
       70-80% of total river networks (Gomi et al., 2002). That these rates are similar could suggest either that the rate of
       transformation was relatively consistent from one location to the other, or that new material was constantly entering the river



system away from the headwaters. The latter is unlikely; however, as from a mass balance perspective the addition of new OM

would be less influential in larger tributaries than in headwaters. Our results suggest that photodegradation is not necessarily limited by the availability of aromatic, HMW, and humic DOM, such that the reactivity of the remaining DOM pool may persist following successive days of exposure. This may explain why the degradation rates obtained for BBR were similar to those seen in BC.

If combined with a reliable field-based method for measuring $CO_2$, the approach presented here could be used to model DOC

degradation's contribution to the production of $CO_2$. This would help to quantify the contribution of DOC decomposition to the total $CO_2$ evasion from rivers (Davidson et al., 2010); which is potentially substantial (OM decomposition is estimated to contribute ~75% of total evasion from the Amazon) but currently poorly constrained (Raymond et al., 2013; Richey et al., 2002). The finding presented here that photodegradation is a more influential process on the complete mineralization of DOC than biodegradation suggests that the former is therefore a more important driver of the degassing of $CO_2$ in tropical

headwaters. The upper limit of DOC photodegradation obtained here (up to 9% over 12 hours) exceeded those observed for water samples from the main stems of larger tropical rivers (Amon and Benner, 1996; Spencer et al., 2009), albeit that these values were obtained with different approaches. This suggests that photodegradation rates, and therefore potentially rates of $CO_2$ degassing, may decease along the river continuum from source to sea. This could be investigated further by conducting incubations along the length of a river network – for example analysing the main stem and mouth of the Essequibo in addition

to BC and BBR – to explore how DOM composition and DOC concentrations change along the river continuum (Cole et al., 2007).

In this study we opted to study daily rates of photo- and biodegradation. This allowed us to study the maximum period of continuous sunlight that river water will experience in 'real world' conditions, and was also in line with estimates of in-stream residence times (though more data are needed regarding tropical headwaters in this regard) (Worrall et al., 2014). However,

our approach could also easily be adapted to include more frequent sampling steps to study hourly degradation rates, and also to further investigate N cycling. From a compositional point of view, this could also shed further light on the complex microbial processes affecting the biopolymer compound group, which was shown here to be particularly reactive; biopolymers contain components important for biota and in particular microbial biofilms (Huber et al., 2011; Her et al., 2002; Flemming et al., 2007). Conversely, longer, multi-day 'dark' incubations may deliver larger, mores discernible and significant changes in DOC

concentrations as a result of microbial processes. This could lead to clearer results regarding the net effects of microbial activity, and could also capture the effects of slower, anaerobic biological processes (e.g., methanogenesis; Stanley et al., 2016). This approach also better reflect scenarios in which flooding leads to standing water and corresponding slower residence times, in addition to downstream sections of the river network, where the flow can become lentic (Junk, 1989). The approach presented here is limited in that it does not quantify POM-DOM interchange during the incubation process – filtering prior to

incubation could circumvent this issue, but would reduce representability, whereas significantly larger samples would be needed to quantify POC.



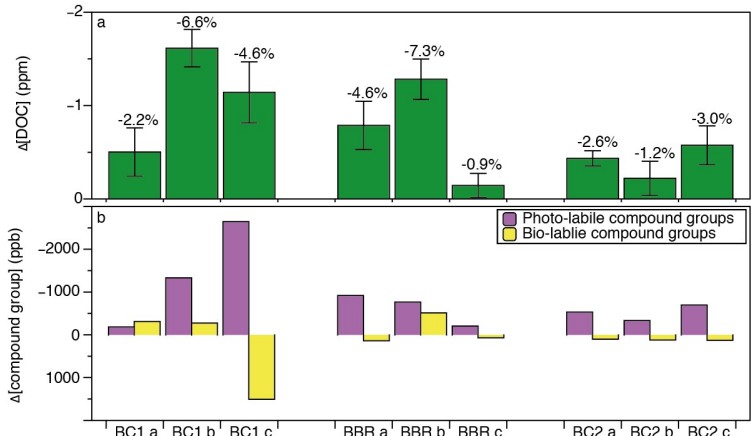

**Figure 6. Overall changes in (a) DOC concentration, and (b) photo-labile and bio-labile compounds across a day-night cycle (i.e. change seen in 'ambient' container over 22 hr incubation); error bars show SD. Note inverted y axes.**

The degree of DOC degradation observed in BC is higher than that observed in headwaters in other climate zones – like a boreal headwater (0.2 km$^2$ catchment) in Sweden (~10% decrease in DOC over 48 hrs (Köhler et al., 2002)) – but is lower than that observed in a temperate peatland headwater (0.2 km$^2$) in the UK (~40% decrease over 24 hrs(Moody and Worrall,

2016)) and two temperate forest streams in Japan (~21-36% over 12-13 hrs (Mostofa et al., 2007, 2005b)). Application of the technique demonstrated here to different climate zones and different river stages could help to quantify how variations in DOC concentrations, environmental conditions, and DOM composition help to modulate degradation.

This study can also address the relative importance of combined photochemical and microbial influences on DOM composition and DOC concentration, compared to other potential influences. Hydrological mobilization at Blackwater Creek has been

shown to drive to variations in DOC one-to-two orders of magnitude larger than those measured in this study, over a shorter timescale (Pereira et al., 2014a); this was also accompanied by greater compositional variability than observed here (Fig. 7b). Further downstream, however, this terrestrial-driven variability may fade, leading to combined photochemical and microbial influences becoming more important. Additionally, their relative importance in tropical headwaters will also likely increase outside of the rainy season, when mobilization of DOM from rainstorms will become less frequent.


## 5 Conclusions

In summary, we reveal that photodegradation is an important process in tropical headwaters – a key component of the global carbon cycle – regarding both the complete mineralization of DOC (and therefore potentially the generation of greenhouse
gases) and the alteration of DOM composition. Daily, the degradation of DOC in tropical headwaters and tributaries occurs at

a similar scale (~1-7%) as in larger tropical rivers (Amon and Benner, 1996; Spencer et al., 2009). Photodegradation has a larger influence on DOC transformation than microbial activity, which had a more varied and inconclusive response. Our study demonstrates that sunlight alone does not drive the extent of photodegradation over daily timescales, rather it appears to be the product of a complex range of influences, possibly including the initial physical properties of the water column and the initial concentration and composition of DOM (Jones et al., 2016; Mostofa et al., 2013a), linking with the surrounding environment

and hydro-chemistry of the river headwater.

**Data availability**

All data are available in the supplementary material (see link below).

**Supplement Link**

**Author contributions**

JFS, RP, TW, JBi, SN, WH, and ST designed the study; JFS, RP, JBi, ST, SN, JBr, and LJ conducted field sampling; JFS, RP, JBi, ST, and WH conducted laboratory analysis; JFS, RP, and TW wrote the manuscript, with additional contributions from JBi, ST, SN, JBr, and LJ.

**Competing interests**

The authors declare that they have no conflict of interest.

**Acknowledgements**

The authors would like to thank Dane Gobin, Raquel Thomas, and the staff of the Iwokrama International Iwokrama International Centre for Rainforest Research and Development (IIC) for their support, advice, and expertise during the planning and implementation of the research program. We would also like to thank Alan MacDonald at the British Geological Survey (BGS) for advice, and for loaning equipment. We would like to dedicate this study to our late friend and colleague, Walter

Hill.





**Financial Support**

This study was supported by funding from IIC, and RP and SNA acknowledge funding from the BGS BUFI scheme. RP acknowledges support from the European Research Council (ERC) under the European Union's Horizon 2020 research and innovation programme (grant agreement No. 949495).

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
