# Peer review of "Unravelling Light and Microbial Activity as Drivers of Organic Matter Transformations in Tropical Headwater Rivers"

_Biogeosciences, 2021_

## Referee Comment (RC1)

Overview:

The authors present a study of DOM reactivity via photodegradation and microbial decomposition of headwaters of the tropical Essequibo River, quantified using bulk DOC and classes derived from Size Exclusion Chromatography (SEC). Incubations were performed in the river using both light and dark control chambers to separate the different effects of photochemistry and biodegradation. The authors attribute the most apparent changes in DOM composition to occur rapidly due to photodegradation with microbial processing resulting in unclear and often inconsistent trends. I appreciate the experimental approach in trying to mimic in situ riverine conditions and the careful use of replicate incubations; however, the manuscript contains serious flaws in the data presentation and interpretations that prevent it from being publishable in its current form. I have done my best to outline areas of improvement and offered suggestions in the comments below.

General Comments:

*Lability:*

My biggest issue with this manuscript is the definition and use of 'photo-labile' and 'bio-labile' molecular categories. From the manuscript it seems like the authors are predefining what 'photo-labile' and 'bio-labile' SEC classes are based on the literature and then tracking their increases and decreases throughout the incubation experiments. This is problematic for several reasons. First, the authors assume that certain SEC compound classes, which are considered not 'photo-labile' in certain studies (Line 263), must therefore by default be 'bio-labile'. Neither the Stubbins et al. (2010) nor Koehler et al. (2012) studies that are cited used SEC to classify DOM compositions, so how can the authors use them to define certain SEC-specific compound classes as being 'bio-labile' (line 265)? Second, from figure 3, LMW N and LMW A both decrease in some of replicates attributed to photochemical effects (even though the authors consider them to be 'bio-labile') and HS and building blocks both decrease in some of replicates attributed to the microbial effect (even though the authors consider them to be 'photo-labile'). These arbitrary classifications are forcing compound classes to be either 'photo-labile' or 'bio-labile' and don't allow for the possibility of DOM to be both or neither. In reality, photo- and bio-lability are complex concepts and DOM compounds are rarely either one or the other. I suggest reading papers such as Bittar et al. (2015a,b) for guidance on how to navigate these classifications.

It would be better to present the compound class data first and use that as a framework for identifying which classes seem to be 'bio-labile' and/or 'photo-labile' in this system rather than trying to force the compounds into these categories beforehand without any justification.

*DON:*

I appreciate the authors' efforts in addressing the organic N composition of DOM in the incubations (section 3.3), but the way it is presented does not make it seem like a relevant part of the manuscript. The introduction does not sufficiently prepare the reader for the importance or context of N cycling in any way, mentioning only that DON is a part of DOM (Line 49). The analysis and treatment of DON is interesting, but on their own the results do not indicate that DON is an important part of the story that the authors are trying to convey, stating that there were no clear trends in C/N ratios and speculating that the changes must be due to other processes without any proper justification or support. In fact, the Kellerman et al. (2015) reference that the authors cite (Line 390) does not even use SEC for their analysis, yet the authors still try to force their SEC-specific compound classes into incompatible frameworks to explain their data. Furthermore, there is very little effort to try to relate the DON results to the trends seen with DOM, given that the authors claim that DON is an important subset of DOM in the introduction. Outside of a single sentence about investigating future N-cycling (Lines 429-431), there are no mentions of DON in the discussion section, conclusion, or even the abstract.

If the authors wish to include DON in the manuscript, I suggest: 1) properly explaining its importance and context in tropical rivers in the introduction section; 2) extensively trim section 3.3 to a single concise paragraph; 3) move figure 5 and any other peripheral details in section 3.3 to the supplemental data; and 4) relate the DON results to the changes in DOM composition in the discussion section.

*Structure and organization:*

There are several instances in the manuscript where sentences unrelated to the topics of the paragraphs are spliced in and derail any sort of flow that begins to develop. There are also areas in the methods and results sections where the authors begin lengthy discussions on the interpretations and implications of their data that are better suited for the discussion section. In both cases, I have done my best to identify these areas in the specific comments below. While I understand that these stylistic conventions may have been made intentionally by the authors, they make the manuscript difficult to read and understand. Fixing the organization will make the main findings clearer for the reader to comprehend.

Specific Comments:

*Abstract*:

Line 16: What does 'supposedly photo-resistant' mean? Why not just say 'photo-resistant' since you already say 'photo-sensitive' right before?

Line 16: 'Microbial activity...bio-labile components'—this sentence is really vague, and I don't think it presents any useful information to the reader.

Lines 18-19: 'Biopolymers…degassing'—These two sentences don't flow with the preceding or proceeding sentences and seem disjointed.

*Introduction:*

Lines 25-29: These sentences jump from $CO_2$ to methane back to $CO_2$ without any reasoning or flow.

Line 43: 'Tropical…emissions'—This sentence seems out of place with the previous and following sentence.

Line 48-49: 'DOM…DON'—it also includes dissolved organic S and organic P. If you're not going to mention them then just cut the DON out of it because it doesn't seem like you discuss DON anywhere else in the introduction. If you are presenting data regarding DON, then please add a section describing its importance in tropical rivers to the introduction.

Line 70: 'Leading to…via respiration'—it can also lead to smaller DOM compounds and byproducts being formed.

Line 80: 'Ignoring' implies that it detects it and doesn't account for it. UV-vis simply can't detect optically invisible DOM.

Line 82: '…difficult to align with quantitative DOC changes…'—Not really, several papers have demonstrated the quantitative nature of FT-ICR MS such as for organic S and organic N (e.g., Poulin et al., 2014; Kurek et al., 2020).

Line 83: I wouldn't call the technique 'novel' if it has been implemented since 2011 with over 1,000 citations of the original paper.

Line 86: '…for example…Arctic river'—The Voss et al. (2015) paper doesn't even use SEC, why even bring it up here?

*Methods:*

Lines 133-136: 'to address…container'—Much of this material seems like it should go in the discussion. Please summarize for the methods.

Lines 183-187: 'Previous studies…of HS'—Much of this material seems like it should go in the results. Please summarize for the methods.

*Results:*

Line 210: I wouldn't say non-aromatic compounds were 'preferentially removed' because overall absorbance did decrease during the incubations, meaning aromatic compounds must have been removed in enough quantity to decrease the absorbance.

Line 213 and throughout: 'four incubations…five of the incubations'—This terminology is misleading and makes it sound like you have 9 different incubation studies. You have 3 incubations each having 3 replicates. I would recommend just referring to them as replicates throughout the manuscript, but if the authors prefer to refer to the replicates as 'incubations', then this terminology needs to be clarified in the Methods section.

Lines 230-251: This section seems like it belongs in the discussion.

Figure 3: This figure is a little misleading. The reader doesn't know the starting concentrations of each DOM group unless they go digging through the supplemental and thus has no context for the concentration changes. I would make figure S2 the main text figure with figure 3 in the supplemental and make the percent changes the main focus, or find a better way to convey the context of the concentration changes to the reader. Also, the caption does not match figure 3. A and B are both photochemical effects in the figure, C and D are labeled as microbial effects. E and F look like initial compositions. Finally, I suggest making an inset for BC2 and possibly BBR, I can't tell what's happening there due to the way the y axes are scaled.

Lines 310-329: This section seems like it belongs in the discussion.

Lines 350-362: This section seems like it belongs in the discussion.

*Discussion:*

Lines 403:405 and figure 6: It seems like you should present this figure before figure 2 as these are the overall changes you observe and figure 2 investigates where all the changes come from (e.g., either microbial or photo-induced).

Lines 409-410: Why would the addition of new OM be less influential in larger tributaries than in headwaters? Large rivers such as the Amazon receive greater OM during periods of high precipitation/discharge which increases the overall DOC concentration and changes the DOM composition (e.g., Seidel et al., 2016).

Lines 410-412: 'Our results suggest…successive days of exposure'—The authors only incubated for less than a day making it difficult to speculate on the long-term kinetics of DOM degradation. Please elaborate and justify this claim with relevant literature.

Lines 418-420: I agree that the authors demonstrated a greater DOC loss in their incubations due to photochemistry, but I would not say that this study suggests that photodegradation is a more important driver of degassing in tropical headwaters. This study omits the sediment where biodegradation also occurs. It also only sampled within 22 hours which likely does not accurately represent the kinetics of biodegradation for this system, as the authors recognize in lines 434-437. There also may be more biolabile sources of DOM upstream that have been mineralized before the sampling sites in this study.

Line 423: 'decease'—I assume the authors meant to write 'decrease'.

Lines 427-429: This is almost an exact repetition of the first sentence in the discussion (Lines 401-403).

Lines 437-438: 'This approach…can become lentic'—I agree that this approach better represents systems with standing water, but I simply don't know if this approach represents the study site since the authors never provide or discuss water residence time and discharge at the time of sampling and the typical conditions for these streams. If water residence time is short at these sites, the incubations likely don't represent the actual conditions as the authors are effectively isolating a parcel of water that would have normally interacted with different light, temperature, sediment, groundwater, and microbial communities over the course of 22 hours.

Lines 438-441: 'The approach…to quantify POC'—I appreciate the authors' recognition of potential DOM-POM interactions, but I would like more of a discussion of the effects of adsorption to DOM in these incubations. Since these incubations are unfiltered, DOM is likely adsorbing to suspended particle and mineral surfaces over time, fractionating the remaining DOM composition. This process can occur rather quickly and drastically change the composition of the remaining DOM compounds in solution (e.g., Lv et al., 2016; Coward et al., 2019). I realize that there are likely fewer iron hydroxides and other minerals at these sites than in the cited studies, so the effects of adsorption are probably not as extreme, but I still think they are worth considering given the changes in DOM classes you observed and are unable to explain. b

Lines 446-450: This may be true, but all these cited experiments have been under different conditions with different methodology making it difficult to compare directly with your results. If the authors wish to compare their findings to these studies, please also discuss how differences in methodology and experimental conditions may have impacted the ranges reported for DOC degradation.

Line 456: 'Fig. 7b'—I do not see figure 7 anywhere in the manuscript or supplemental info.

*Conclusion:*

Lines 465-466: 'Photodegradation has a larger influence on DOC transformation than microbial activity, which had a more varied and inconclusive response.'—I think this is only true based on

the timescales and conditions of this study but may or may not be true of the natural system. Please clarify that these results are specific to the experimental conditions of the study.

References:

Bittar, T. B., Stubbins, A., Vieira, A. A., & Mopper, K. (2015a). Characterization and photodegradation of dissolved organic matter (DOM) from a tropical lake and its dominant primary producer, the cyanobacteria Microcystis aeruginosa. *Marine Chemistry*, *177*, 205-217.

Bittar, T. B., Vieira, A. A., Stubbins, A., & Mopper, K. (2015b). Competition between photochemical and biological degradation of dissolved organic matter from the cyanobacteria Microcystis aeruginosa. *Limnology and Oceanography*, *60*(4), 1172-1194.

Coward, E. K., Ohno, T., & Sparks, D. L. (2018). Direct evidence for temporal molecular fractionation of dissolved organic matter at the iron oxyhydroxide interface. *Environmental science & technology*, *53*(2), 642-650.

Kellerman, A. M., Kothawala, D. N., Dittmar, T., & Tranvik, L. J. (2015). Persistence of dissolved organic matter in lakes related to its molecular characteristics. *Nature Geoscience*, *8*(6), 454-457.

Koehler, B., Von Wachenfeldt, E., Kothawala, D., & Tranvik, L. J. (2012). Reactivity continuum of dissolved organic carbon decomposition in lake water. *Journal of Geophysical Research: Biogeosciences*, *117*(G1).

Kurek, M. R., Poulin, B. A., McKenna, A. M., & Spencer, R. G. (2020). Deciphering Dissolved Organic Matter: Ionization, Dopant, and Fragmentation Insights via Fourier Transform-Ion Cyclotron Resonance Mass Spectrometry. *Environmental Science & Technology*, *54*(24), 16249-16259.

Lv, J., Zhang, S., Wang, S., Luo, L., Cao, D., & Christie, P. (2016). Molecular-scale investigation with ESI-FT-ICR-MS on fractionation of dissolved organic matter induced by adsorption on iron oxyhydroxides. *Environmental science & technology*, *50*(5), 2328-2336.

Poulin, B. A., Ryan, J. N., Nagy, K. L., Stubbins, A., Dittmar, T., Orem, W., ... & Aiken, G. R. (2017). Spatial dependence of reduced sulfur in Everglades dissolved organic matter controlled by sulfate enrichment. *Environmental science & technology*, *51*(7), 3630-3639.

Seidel, M., Dittmar, T., Ward, N. D., Krusche, A. V., Richey, J. E., Yager, P. L., & Medeiros, P. M. (2016). Seasonal and spatial variability of dissolved organic matter composition in the lower Amazon River. *Biogeochemistry*, *131*(3), 281-302.

Stubbins, A., Spencer, R. G., Chen, H., Hatcher, P. G., Mopper, K., Hernes, P. J., ... & Six, J. (2010). Illuminated darkness: Molecular signatures of Congo River dissolved organic matter and its photochemical alteration as revealed by ultrahigh precision mass spectrometry. *Limnology and Oceanography*, *55*(4), 1467-1477.

---

## Author Comment (AC1)

**Lyell Centre for Earth and Marine Sciences**

Dr James Spray
The Lyell Centre
Heriot-Watt University
Research Avenue South
Edinburgh
EH14 4AS, UK
Jfs8@hw.ac.uk
Tel: +44 131 451 3537

16th July, 2021

Dear Anonymous Reviewer 1,

Thank you for your review of our manuscript 'Unravelling light and Microbial Activity as Drivers of Organic Matter Transformations in Tropical Headwater Rivers', authored by Spray et al., which we submitted for publication in *Biogeosciences*. Please find below our responses to each of your comments in blue text.

Yours Faithfully,

Dr James Spray and Dr Ryan Pereira,

On behalf of all co-authors (Thomas Wagner, Juliane Bischoff, Sara Trojahn, Sevda Norouzi, Walter Hill, Julian Brasche, and Leroy James)

**Overview:**

The authors present a study of DOM reactivity via photodegradation and microbial decomposition of headwaters of the tropical Essequibo River, quantified using bulk DOC and classes derived from Size Exclusion Chromatography (SEC). Incubations were performed in the river using both light and dark control chambers to separate the different effects of photochemistry and biodegradation. The authors attribute the most apparent changes in DOM composition to occur rapidly due to photodegradation with microbial processing resulting in unclear and often inconsistent trends. I appreciate the experimental approach in trying to mimic in situ riverine conditions and the careful use of replicate incubations; however, the manuscript contains serious flaws in the data presentation and interpretations that prevent it from being publishable in its current form. I have done my best to outline areas of improvement and offered suggestions in the comments below.

We thank the reviewer for their considered comments and suggestions; we agree that the unique strength of our approach is in that it intends to mimic DOM transformations in remote tropical headwater rivers in-situ under 'real' environmental conditions, and to analyse arising compositional DOM changes using SEC. We actually believe that the core of the criticism stems from the complex nature of the data presented, reflecting the wide range of methodologies and factors used to characterize DOM composition, the pioneering and therefore incomplete set-up of the field experiments, and the associated challenges of distilling a generic new process understanding. To remedy this, we have carefully re-analysed our presentation strategy and propose modifications to the text, figures, and manuscript structure. We outline the reasoning and proposed changes in our response to the reviewer's general and specific comments below. We would like to thank the reviewer for their criticism; it helped us to reframe and refocus the manuscript by focusing on its strengths and novelty, while also better addressing its limitations to better highlight the novelty of our study.

**General Comments:**

**Lability:** My biggest issue with this manuscript is the definition and use of 'photo-labile' and 'bio-labile' molecular categories. From the manuscript it seems like the authors are predefining what 'photo-labile' and 'bio-labile' SEC classes are based on the literature and then tracking their increases and decreases throughout the incubation experiments. This is problematic for several reasons. First, the authors assume that certain SEC compound classes, which are considered not 'photo-labile' in certain studies (Line 263), must therefore by default be 'bio-labile'. Neither the Stubbins et al. (2010) nor Koehler et al. (2012) studies that are cited used SEC to classify DOM compositions, so how can the authors use them to define certain SEC-specific compound classes as being 'bio-labile' (line 265)? Second, from figure 3, LMW N and LMW A both decrease in some of replicates attributed to photochemical effects (even though the authors consider them to be 'bio-labile') and HS and building blocks both decrease in some of replicates attributed to the microbial effect (even though the authors consider them to be 'photo-labile'). These arbitrary classifications are forcing compound classes to be either 'photo-labile' or 'bio-labile' and don't allow for the possibility of DOM to be both or neither. In reality, photo-and bio-lability are complex concepts and DOM compounds are rarely either one or the other. I suggest reading papers such as Bittar et al. (2015a, b) for guidance on how to navigate these classifications. It would be better to present the compound class data first and use that as a framework for identifying which classes seem to be 'bio-labile' and/or 'photo-labile' in this system rather than trying to force the compounds into these categories beforehand without any justification.

Regarding the reviewer's first point, to our knowledge no previous study has used SEC to specifically study the photo- and/or bio-degradation of DOM. Our main point is that the observations obtained from each method are complimentary and synergistic, irrespective of methodological differences between SEC and other methods used in published studies. This concept should also apply to trends in DOM behaviour pertaining to differences in molecular weight and to differences in their optical properties. As we are all aware, each methodology has strengths and limitations and directly

comparing individual variables from different techniques and experimental set-ups remain challenging, and for specific combinations maybe not be justified. This principle also applies to our study. We argue, however, that the compositional groups identified by SEC, delineated by molecular weight (proportional to elution time) and UV absorbance, are appropriate to consider trends of photo- or bio-lability and resulting DOM properties (albeit measured by different techniques). Combined, these enable us to cautiously hypothesize whether each compositional group might be more photo- or bio-labile. We are fully aware that there are uncertainties involved in making these hypotheses as for every other analytical approach.

Regarding the reviewer's second point, our intention is not to predefine but rather to explore how the DOM component groups identified by SEC could potentially be grouped to reflect their lability or recalcitrance to either photochemistry or microbial action and build testable hypothesis around this analysis. As the reviewer correctly states, there is indeed evidence for overlap between bio-labile and photo-labile DOM pools, as highlighted by the findings of Bittar et al. (2015b), which revealed an overlap of up to 15%. In a different study, Amado (2006) concluded that photo- and biodegradation were complementary in Amazon clear waters, degrading different fractions of the DOM pool. Despite being highly relevant regarding the key question, both studies are strikingly different to the approach we have selected for our study. Bittar et al focuses on characterizing extracellular and intracellular DOM pools, while Amando analyses white waters and lake waters from the Amazon basin. We analyse in situ processes in a small black water headwater and its larger immediate receiving waters. We are therefore very cautious to expect that the results from our and other published studies can be directly compared, though all studies target at the same (complex) phenomenon, but from very different perspectives.

Our approach is intended to be a starting point (using a novel experimental set-up and analytical technique in a largely understudied (but arguably important) sub-environment, small headwaters) with which to address the compositional DOM data. This central aspect will be stronger emphasised in our revisions. We plan to go into greater detail in our results and discussion by addressing the responses of individual component groups, and recognizing their apparent reactivity to both processes (e.g. "…HS was removed by microbial activity in five of the incubations, including BC1-c and BBR-c, at a similar magnitude to that of photodegradation. It appears therefore, that even HMW molecules within this compound group are bio-available…" (Lines 305-306)). Our proposed restructuring of the results and discussion section (outlined below) will be refocused and collated within the new section 4.2 that targets the complexity/overlap of photo- and microbial influences.

To firmly address these two significant points, we suggest adding the following section to the introduction:

"The compositional groups identified through this technique are delineated based on their optical properties (i.e. their UV absorbance at 254 nm, a proxy for CDOM) and their molecular weight (Huber et al., 2011). To the authors' knowledge, SEC has not yet been applied to study compositional changes in response to photo- and biodegradation. However, as highlighted above, many studies using a variety of other approaches have characterized the relative photo- and or bio-lability of DOM. While there are limitations in our ability to directly compare to other methodological approaches, it is possible to relate the potential reactivity of SEC compositional groups to other work, based upon previous characterisations of their properties (Huber et al., 2011). For example, Spencer et al., (2009) used UV-visible spectrophotometry to demonstrate that CDOM is photodegraded at a higher rate than bulk DOC. Thus, UV-amenable component groups identified by SEC could be relatively photo-labile. Conversely, several studies with varying methodologies have shown that the biolability of DOM correlates positively to lower molecular weights and negatively to its aromaticity (Kaplan and Cory, 2016, and references therein), suggesting that LMW, optically invisible SEC components could be relatively bio-labile. Though bio- and photo-labile DOM pools may or may not overlap in different

environments (Amado et al., 2006; Bittar et al., 2015b), these hypotheses represent a starting point with which to apply SEC to the investigation of DOM transformations."

**DON:** I appreciate the authors' efforts in addressing the organic N composition of DOM in the incubations (section 3.3), but the way it is presented does not make it seem like a relevant part of the manuscript. The introduction does not sufficiently prepare the reader for the importance or context of N cycling in any way, mentioning only that DON is a part of DOM (Line 49). The analysis and treatment of DON is interesting, but on their own the results do not indicate that DON is an important part of the story that the authors are trying to convey, stating that there were no clear trends in C/N ratios and speculating that the changes must be due to other processes without any proper justification or support. In fact, the Kellerman et al. (2015) reference that the authors cite (Line 390) does not even use SEC for their analysis, yet the authors still try to force their SEC-specific compound classes into incompatible frameworks to explain their data. Furthermore, there is very little effort to try to relate the DON results to the trends seen with DOM, given that the authors claim that DON is an important subset of DOM in the introduction. Outside of a single sentence about investigating future N-cycling (Lines 429-431), there are no mentions of DON in the discussion section, conclusion, or even the abstract. If the authors wish to include DON in the manuscript, I suggest: 1) properly explaining its importance and context in tropical rivers in the introduction section; 2) extensively trim section 3.3 to a single concise paragraph; 3) move figure 5 and any other peripheral details in section 3.3 to the supplemental data; and 4) relate the DON results to the changes in DOM composition in the discussion section.

We refer to our response to the previous point and propose the following alterations aligned with the suggestions by the reviewer:

1) Adding the following dedicated section to the introduction:

"In addition to DOC, the DOM pool also encompasses Dissolved Organic Nitrogen (DON), among other components. It is important to consider the role of DON as well as DOC when exploring the response of DOM to microbial and photochemical influences. In addition to influencing net DOC concentrations and DOM compositions, photodegradation can also break down DON, leading to the formation of Dissolved Inorganic Nitrogen (DIN) in the form of nitrate and ammonium, which are critical for biological productivity (Zepp, 2003). From a microbial perspective, it has also been suggested that LMW, N-containing DOM in streams is relatively biolabile (Kaplan and Cory, 2016). Furthermore, the availability of DIN has been shown to limit the biological metabolism of DOM, whereas DON has been shown to be utilized by bacteria at a higher rate than DOC, affecting the rate at which they are biologically processed (Wiegner and Seitzinger, 2001) and thereby altering the C/N ratio.

2) Trimming the existing section 3.3 as follows (with Figure 5 being moved to Supplemental information):

"As highlighted in Section 2.2, LC-OCD-OND permits the quantification of DON in the HS and biopolymer component groups, and of DIN (ammonium and nitrate). We note that the initial availability of DIN measured in our samples cannot not explain the variation in the microbial-driven changes in DOC (Fig. 2a, Table S4). The initial C/N ratios of the HS fraction were relatively high (Fig. S3), suggesting a terrigenous source for this fraction, in agreement with the HS diagram (Fig. 4; Perdue and Koprivnjak, 2007). During incubation, neither the biopolymer nor HS component groups (Fig. S3) displayed a clear trend in C/N ratios, for either photochemical or microbial influences.

Furthermore, changes in the C/N ratios from neither component group correlated with observed changes in ammonium or nitrate (Fig. S3)."

3) Adding the following paragraph to section 4.4 of the discussion (see proposed layout below):

"Though we demonstrated the photo-induced degradation of HS, this was not accompanied by the production of DIN photoproducts (ammonium or nitrate; Fig. S3). Thus, despite the prevalence of HS in tropical headwaters, their breakdown does not appear to be a significant source of DIN; N-dynamics in tropical headwaters may therefore be dominated by DON sourced from elsewhere (e.g. building blocks, LMW neutrals, or LMW acids, which LC-OCD-OND cannot quantify). Previous studies, albeit with differing techniques, have shown that N-containing compounds are more associated with LMW, aliphatic molecules (Kaplan and Cory, 2016; Kellerman et al., 2015), lending weight to the idea that N stability in the DOM pool may be controlled by these compounds; tandem SEC and FT-ICR-MS analyses could help to better reveal this. It is also possible that, (photo-induced) microbial scavenging may have outstripped the production of DIN; our study set-up captured significant degradation of DOC on daily timescales, but photochemically and/or microbial-driven N cycling between organic and inorganic pools may have occurred on timescales too rapid to be tracked by our approach (Zepp, 2003). This concept could be tested in future by filtering or otherwise altering water prior to incubation to remove bacteria, combined with hourly sampling intervals."

Regarding the methodological differences between SEC and FT-ICR-MS, optical techniques, etc., as outlined above there is still a possibility for the information that is obtained from each method to be complimentary and synergistic – particularly regarding trends in DOM behaviour pertaining to differences in molecular weight, and to differences in their optical properties.

**Structure and organization:** There are several instances in the manuscript where sentences unrelated to the topics of the paragraphs are spliced in and derail any sort of flow that begins to develop. There are also areas in the methods and results sections where the authors begin lengthy discussions on the interpretations and implications of their data that are better suited for the discussion section. In both cases, I have done my best to identify these areas in the specific comments below. While I understand that these stylistic conventions may have been made intentionally by the authors, they make the manuscript difficult to read and understand. Fixing the organization will make the main findings clearer for the reader to comprehend.

We propose to better delineate the results and discussions sections as follows, with interpretations being moved to the latter:

3. Results

3.1. Photochemical and microbial-driven changes in DOC concentrations

3.2. Photochemical and microbial driven changes in DOM composition

3.3. Organic Nitrogen Dynamics of DOM

3.4. Combined photochemical and microbial-driven changes over a day-night cycle

4. Discussion

4.1. Factors influencing photochemical and microbial-driven changes in DOC concentrations

4.2. Photo- and bio-lability of DOM component groups

4.3. Advantages and limitations of in-situ measurements

4.4. Implications and further research

Likewise, the justifications/elaborations of the methodological approach will be moved from the methods to the discussion section (Section 4.3), with the existing methods section being streamlined.

**Specific Comments:**

**Abstract:**

*Line 16:* What does 'supposedly photo-resistant' mean? Why not just say 'photo-resistant' since you already say 'photo-sensitive' right before? Line 16: 'Microbial activity...bio-labile components'—this sentence is really vague, and I don't think it presents any useful information to the reader.

*Lines 18-19:* 'Biopolymers...degassing'—these two sentences don't flow with the preceding or proceeding sentences and seem disjointed.

In light of these points, we propose the following clarifications to the abstract:

"……on average sunlight oxidised 5% of dissolved organic carbon (DOC) over 12 hours, at rates higher than or comparable to larger tropical rivers. Larger, ultraviolet (UV)-absorbing DOM components were removed, whereas optically invisible lower molecular weight (LMW) components showed a variable response. Microbial activity had varying, less clear influences on DOC concentrations or DOM compositional groups, with no preferential consumption of LMW components; biopolymers were particularly reactive to both processes. Overall, we show sunlight has a greater potential to mineralise tropical headwater DOM than microbial processes and thus potentially influence degassing."

**Introduction:**

*Lines 25-29:* These sentences jump from $CO_2$ to methane back to $CO_2$ without any reasoning or flow.

We propose the following revision, which groups $CO_2$ and methane together as greenhouse gases:

"However, only 0.9 Pg C yr$^{-1}$ is estimated to reach the ocean; with significant volumes of this flux being mineralised and released as greenhouse gases such as carbon dioxide ($CO_2$) and methane ($CH_4$). Approximately 3.9 Pg C yr$^{-1}$ is emitted to the atmosphere as $CO_2$ (Battin et al., 2008; Cole et al., 2007; Drake et al., 2018), and wetlands (in particular tropical wetlands) represent the largest non-anthropogenic source of $CH_4$, comprising 20-30% of total $CH_4$ emissions (Nisbet et al., 2014; Bousquet et al., 2011)."

*Line 43:* 'Tropical...emissions'—this sentence seems out of place with the previous and following sentence.

We propose the following revision, which we believe flows more naturally:

"Indeed, the annual flux of dissolved organic carbon (DOC) from tropical rivers to the ocean accounts for ~59% of global riverine DOC flux (Huang et al., 2012), and the Amazon River alone emits 470 Tg C yr$^{-1}$ of $CO_2$ to the atmosphere. Up to 75% of these emissions are estimated to derive from OM of near-channel origin, which has been mobilized to, and mineralized within, the aquatic zone (Davidson et al., 2010; Richey et al., 2002)."

*Line 48-49:* 'DOM...DON'—it also includes dissolved organic S and organic P. If you're not going to mention them then just cut the DON out of it because it doesn't seem like you discuss DON anywhere else in the introduction. If you are presenting data regarding DON, then please add a section describing its importance in tropical rivers to the introduction.

As discussed above, we are proposing to rework the DON references as suggested. As such, we propose removing the mention of DON on line 48 and instead adding a dedicated DON section to the introduction, as outlined above.

*Line 70:* 'Leading to...via respiration'—it can also lead to smaller DOM compounds and by products being formed.

We propose the following alteration:

"…leading to DOM mineralization and $CO_2$ production via respiration, as well as to alterations in DOM composition and the formation of by products (Cory et al., 2014)."

*Line 80:* 'Ignoring' implies that it detects it and doesn't account for it. UV-vis simply can't detect optically invisible DOM.

We propose altering 'ignores' to 'does not capture'

*Line 82:* '...difficult to align with quantitative DOC changes...'—Not really, several papers have demonstrated the quantitative nature of FT-ICR MS such as for organic S and organic N (e.g., Poulin et al., 2014; Kurek et al., 2020).

We propose the following alteration, referring to the issues of using FT-ICR MS regarding solid phase extraction:

"…but due to the process of solid phase extraction cannot cover the full size range of DOM molecules; it excludes biopolymers, a key DOM component."

*Line 83:* I wouldn't call the technique 'novel' if it has been implemented since 2011 with over 1,000 citations of the original paper.

We propose removing the word 'novel'.

*Line 86:* '...for example...Arctic river'—The Voss et al. (2015) paper doesn't even use SEC, why even bring it up here?

Our intention was to refer only to the fact that Voss et al. had also shown rapid variability in composition and flux, rather than implying this study had the same methodology as Pereira et al. (2014). However, we recognise that this point is already made earlier in the introduction section (Line 50), and so have removed the reference to it highlighted by the reviewer.

**Methods:**

*Lines 133-136:* 'to address...container'—Much of this material seems like it should go in the discussion. Please summarize for the methods.

*Lines 183-187:* 'Previous studies...of HS'—Much of this material seems like it should go in the results. Please summarize for the methods.

See the proposed structural reorganization above.

**Results:**

*Line 210:* I wouldn't say non-aromatic compounds were 'preferentially removed' because overall absorbance did decrease during the incubations, meaning aromatic compounds must have been removed in enough quantity to decrease the absorbance.

We believe that this issue arises from a confusion/conflation between aromaticity as indicated by $SUVA_{254}$ (i.e. DOC normalised) and colour/absorbance (as indicated by absorbance at 254 nm) – the difference in the responses of $SUVA_{254}$ and absorbance at 254 nm suggests that these two properties

are not perfectly aligned; DOC concentrations may therefore be a controlling influence in the interpretation of SUVA. Thus, it is correct to say that coloured compounds were removed as the reviewer highlights, but they appear to have been removed at a slower rate than total DOC. We have made the following edits to clarify this point:

"Interestingly, however, the specific UV absorbance at 254 nm ($SUVA_{254}$) in all but one incubation increased due to photochemical influences (~2-5%, excluding BC1-a; Table 1). It appears then that though photobleaching occurred, $UV_{254}$ decreased at a slower rate than the DOC concentration, suggesting that CDOM compounds were not removed at a faster rate than the total DOC pool."

*Line 213 and throughout:* 'four incubations...five of the incubations'—this terminology is misleading and makes it sound like you have 9 different incubation studies. You have 3 incubations each having 3 replicates. I would recommend just referring to them as replicates throughout the manuscript, but if the authors prefer to refer to the replicates as 'incubations', then this terminology needs to be clarified in the Methods section.

We propose to re-word uses of "incubation" to make distinctions between replicates at each site and different incubations at different sites. We will also clarify this terminology in the methods section.

*Lines 230-251:* This section seems like it belongs in the discussion.

See the proposed structural reorganization above.

*Figure 3:* This figure is a little misleading. The reader doesn't know the starting concentrations of each DOM group unless they go digging through the supplemental and thus has no context for the concentration changes. I would make figure S2 the main text figure with figure 3 in the supplemental and make the percent changes the main focus, or find a better way to convey the context of the concentration changes to the reader. Also, the caption does not match figure 3. A and B are both photochemical effects in the figure, C and D are labelled as microbial effects. E and F look like initial compositions. Finally, I suggest making an inset for BC2 and possibly BBR, I can't tell what's happening there due to the way the y axes are scaled.

We propose to alter the figure caption to correctly describe the data shown in each panel, as follows:

"**Figure 3. a) Photochemistry-induced, and c) microbial-induced changes in the concentrations of photo-labile (HS, protein-based biopolymers and building blocks) and bio-labile compounds (non-protein-based biopolymers, LMW neutrals, and LMW acids) over 12 h; e) initial concentrations of photo-labile and bio-labile compound groups. B) Photochemistry-induced, and d) microbial-induced changes in the concentration of each DOM component over 12 h; e & f) DOM compositions of initial samples for each incubation. Note inverted y axes for (a-d).**"

We propose showing the initial concentrations of each component DOM group instead of percentage changes, as shown below (e, f), with figure S2 then showing percentages. We also propose adding insets for BC2 in panels b and d, as suggested.

[Figure]

*Lines 310-329:* This section seems like it belongs in the discussion.

*Lines 350-362:* This section seems like it belongs in the discussion.

See the proposed structural reorganization above.

**Discussion:**

Lines 403:405 and figure 6: It seems like you should present this figure before figure 2 as these are the overall changes you observe and figure 2 investigates where all the changes come from (e.g., either microbial or photo-induced).

Our rationale for placing figure 6 (now figure 5 as the DON figure has been moved to supplementary material- see above) in its original position is that it shows the combined effects (i.e. ambient response) of river water over a day-night cycle, not over 12 hours (the timescale shown in the current figure 2); we view this as secondary to the main aims of exploring photochemical vs. microbial influences on DOC concentration and compositional changes. Furthermore, this figure presents compositional changes (we have now expanded this to discuss group specific changes, see panel c), so putting it ahead of figure 2 would require restructuring to explain compositional changes. We believe that our proposed organisation of the results and discussion helps to alleviate any perceived lack of flow.

[Figure]

**"Figure 5. Overall changes in (a) DOC concentration, (b) photo-labile and bio-labile compounds across a day-night cycle, and (c) DOM component groups (i.e. change seen in 'ambient' container over 22 hr incubation); error bars show SD. Note inverted y axes."**

Lines 409-410: Why would the addition of new OM be less influential in larger tributaries than in headwaters? Large rivers such as the Amazon receive greater OM during periods of high precipitation/discharge which increases the overall DOC concentration and changes the DOM composition (e.g., Seidel et al., 2016).

We propose the following edit: "…That these rates are similar could suggest that the rate of transformation was relatively consistent from one location to the other."

Lines 410-412: 'Our results suggest...successive days of exposure'—the authors only incubated for less than a day making it difficult to speculate on the long-term kinetics of DOM degradation. Please elaborate and justify this claim with relevant literature.

We propose rewording this as follows to clarify and show the need for further research:

"Our results do not cover longer time periods, however, they suggest that photodegradation is not necessarily limited to aromatic, HMW, and humic DOM, such that the photoreactivity of the remaining DOM pool may potentially persist if these were to be depleted over successive days of exposure, leading to similar rates of degradation downstream. There is indirect support for our assumption. The mean degradation observed in our study (5%) is comparable with those of a previous study on river water from the main stems of the Rio Negro, which reported DOC photodegradation of ~5% in unfiltered river water after ten hours of exposure to natural sunlight (Amon and Benner, 1996). It possibly exceeds that observed in the Congo River, which showed ~10% photodegradation of DOC over 48 hours of continuous exposure to a solar simulator (the intensity of which likely exceeds natural sunlight; Spencer et al., 2009). Direct comparison with the Rio Negro or our data is difficult however, given that river water in the Congo study was filtered and poisoned prior to incubation. Further investigation is needed, therefore, to determine whether photodegradation rates, and therefore potentially rates of $CO_2$ degassing, change along the river continuum from source to sea. This could be achieved by conducting incubations along the length of a river network – for example analysing the main stem and mouth of the Essequibo in addition to BC and BBR – to explore how DOM composition and DOC concentrations change along the river continuum (Cole et al., 2007)."

Lines 418-420: I agree that the authors demonstrated a greater DOC loss in their incubations due to photochemistry, but I would not say that this study suggests that photodegradation is a more important driver of degassing in tropical headwaters. This study omits the sediment where biodegradation also occurs. It also only sampled within 22 hours which likely does not accurately represent the kinetics of biodegradation for this system, as the authors recognize in lines 434-437. There also may be more biolabile sources of DOM upstream that have been mineralized before the sampling sites in this study.

We propose the following clarification:

"The findings presented here suggest that photodegradation has a higher potential for in-stream DOC removal than biodegradation, suggesting that the former may therefore be a more important driver of the degassing of $CO_2$ in tropical headwaters, at least on daily timescales. However, further exploration is needed regarding microbial degradation processes hosted within sediment, which were not considered in the present study. "

As discussed in the response to the reviewer's comment regarding residence times below, we believe the timeframe of the incubation is appropriate for quantifying the reactivity of processes in headwaters. As the reviewer highlights, we already discuss the need to do longer incubations to better quantify microbial processes.

Line 423: 'decease'—I assume the authors meant to write 'decrease'.

This assumption is correct and we will revise this accordingly.

Lines 427-429: This is almost an exact repetition of the first sentence in the discussion (Lines 401-403).

We will remove the first occurrence of this sentence (Lines 401-403). The first occurrence of this sentence pertains to introduce the data shown in Figure 6 (now Fig. 5), but as discussed above we are now proposing to move this figure into the new results section and re-write the text accordingly.

Lines 437-438: 'This approach...can become lentic'—I agree that this approach better represents systems with standing water, but I simply don't know if this approach represents the study site since the authors never provide or discuss water residence time and discharge at the time of sampling and the typical conditions for these streams. If water residence time is short at these sites, the incubations likely don't represent the actual conditions as the authors are effectively isolating a parcel of water

that would have normally interacted with different light, temperature, sediment, groundwater, and microbial communities over the course of 22 hours.

We propose the following clarification to better characterize the typical hydrological conditions of the streams, building on our original point on Line 429 regarding the suitability of the incubation time:

"Though no dedicated studies of residence time have been conducted at either study site, the mean velocity measured near to Blackwater creek sites BC1 and BC2 across a period of one month (spanning all incubations) was ~0.29 m/s (max = 0.52 m/s; min 0.05 m/s). Given the stream length of ~6 km, we estimate a mean travel time for the catchment of ~57 h (min =~31 h; max = ~ 14 days). A study of a different tropical stream catchment ~ 1.5x bigger than Blackwater creek determined a mean transit time during the wet season of 15 days (Birkel et al., 2016), though clearly there are differences in hydrology, topography, etc. between the two catchments, making any direct connections/comparisons most difficult."

Lines 438-441: 'The approach...to quantify POC'—I appreciate the authors' recognition of potential DOM-POM interactions, but I would like more of a discussion of the effects of adsorption to DOM in these incubations. Since these incubations are unfiltered, DOM is likely adsorbing to suspended particle and mineral surfaces over time, fractionating the remaining DOM composition. This process can occur rather quickly and drastically change the composition of the remaining DOM compounds in solution (e.g., Lv et al., 2016; Coward et al., 2019). I realize that there are likely fewer iron hydroxides and other minerals at these sites than in the cited studies, so the effects of adsorption are probably not as extreme, but I still think they are worth considering given the changes in DOM classes you observed and are unable to explain.

We are wary of overly inflating this section of the discussion without appropriate results, but we are comfortable to propose the following expansion of our discussion to highlight the point that, as stated by the reviewer, in-incubation adsorption to suspended particles could have led to some internal DOM fractionation:

"A further consideration is that adsorption to iron hydroxides within the water column could have fractionated the DOM pool to some extent; studies have shown that aromatics and polyphenols are more likely to adsorb and become stabilized, whereas aliphatic compounds preferentially remain in solution and are more easily destabilized once adsorbed (Coward et al., 2018).The lack of appropriate data from our in situ experiments, however, does not enable us to take this discussion forward."

Lines 446-450: This may be true, but all these cited experiments have been under different conditions with different methodology making it difficult to compare directly with your results. If the authors wish to compare their findings to these studies, please also discuss how differences in methodology and experimental conditions may have impacted the ranges reported for DOC degradation.

We propose the following alteration:

"The degree of DOC degradation observed in BC is higher than that observed in headwaters in other climate zones – like …….. These comparisons are hampered however by the lack of a consistent methodology used to quantify degradation in each case. Application of the technique demonstrated here to different climate zones and different river stages could therefore help to quantify how variations in DOC concentrations, environmental conditions, and DOM composition help to modulate degradation; whereas companion lab- and field-based incubation approaches conducted at the same sites could help to inform on differences in methodological approaches. Overall, there is a need to derive consistency across approaches."

Line 456: 'Fig. 7b'—I do not see figure 7 anywhere in the manuscript or supplemental info.

This should read Fig. 6 (now Fig. 5 given removal of DON figure to Supplementary info); we propose to correct this accordingly.

**Conclusion:**

Lines 465-466: 'Photodegradation has a larger influence on DOC transformation than microbial activity, which had a more varied and inconclusive response.'—I think this is only true based on the timescales and conditions of this study but may or may not be true of the natural system. Please clarify that these results are specific to the experimental conditions of the study.

We think that in the context of the paragraph in which this statement appears, it is obvious and clear that we are discussing the study location (there are several occurrences of 'tropical headwaters') and the timescale ('Daily, the degradation of…' (previous sentence) and '…over daily timescales…' (following sentence)). Furthermore, one of the points of our study, as we attempted to outline in our introduction, was that smaller spatial and temporal scales (i.e. headwaters, hours-days) need to be further investigated to better understand the natural system. We propose to more clearly stress this point in the conclusion.

**References:**

Amado, A. M., Farjalla, V. F., Esteves, F. D. A., Bozelli, R. L., Roland, F., & Enrich-Prast, A. (2006). Complementary pathways of dissolved organic carbon removal pathways in clear-water Amazonian ecosystems: photochemical degradation and bacterial uptake. *FEMS Microbiology Ecology*, *56*(1), 8-17.

Amon, R. M. W. and Benner, R.: Photochemical and microbial consumption of dissolved organic carbon and dissolved oxygen in the Amazon River system, Geochim. Cosmochim. Acta, 60, 1783–1792, https://doi.org/10.1016/0016-7037(96)00055-5, 1996.

Birkel, C., Geris, J., Molina, M. J., Mendez, C., Arce, R., Dick, J., ... & Soulsby, C. (2016). Hydroclimatic controls on non-stationary stream water ages in humid tropical catchments. Journal of Hydrology, 542, 231-240.

Bittar, T. B., Stubbins, A., Vieira, A. A., & Mopper, K. (2015a). Characterization and photodegradation of dissolved organic matter (DOM) from a tropical lake and its dominant primary producer, the cyanobacteria Microcystis aeruginosa. Marine Chemistry, 177, 205-217.

Bittar, T. B., Vieira, A. A., Stubbins, A., & Mopper, K. (2015b). Competition between photochemical and biological degradation of dissolved organic matter from the cyanobacteria Microcystis aeruginosa. Limnology and Oceanography, 60(4), 1172-1194.

Coward, E. K., Ohno, T., & Sparks, D. L. (2018). Direct evidence for temporal molecular fractionation of dissolved organic matter at the iron oxyhydroxide interface. Environmental science & technology, 53(2), 642-650.

Kaplan, L. A., & Cory, R. M. (2016). Dissolved organic matter in stream ecosystems: forms, functions, and fluxes of watershed Tea. In *Stream ecosystems in a changing environment* (pp. 241-320). Academic Press.

Kellerman, A. M., Kothawala, D. N., Dittmar, T., & Tranvik, L. J. (2015). Persistence of dissolved organic matter in lakes related to its molecular characteristics. Nature Geoscience, 8(6), 454-457.

Koehler, B., Von Wachenfeldt, E., Kothawala, D., & Tranvik, L. J. (2012). Reactivity continuum of dissolved organic carbon decomposition in lake water. Journal of Geophysical Research: Biogeosciences, 117(G1).

Kurek, M. R., Poulin, B. A., McKenna, A. M., & Spencer, R. G. (2020). Deciphering Dissolved Organic Matter: Ionization, Dopant, and Fragmentation Insights via Fourier Transform-Ion Cyclotron Resonance Mass Spectrometry. Environmental Science & Technology, 54(24), 16249-16259.

Lv, J., Zhang, S., Wang, S., Luo, L., Cao, D., & Christie, P. (2016). Molecular-scale investigation with ESI-FT-ICR-MS on fractionation of dissolved organic matter induced by adsorption on iron oxyhydroxides. Environmental science & technology, 50(5), 2328-2336.

Perdue, E. M., & Koprivnjak, J. F. (2007). Using the C/N ratio to estimate terrigenous inputs of organic matter to aquatic environments. *Estuarine, Coastal and Shelf Science*, *73*(1-2), 65-72.

Poulin, B. A., Ryan, J. N., Nagy, K. L., Stubbins, A., Dittmar, T., Orem, W., ... & Aiken, G. R. (2017). Spatial dependence of reduced sulfur in Everglades dissolved organic matter controlled by sulfate enrichment. Environmental science & technology, 51(7), 3630-3639.

Seidel, M., Dittmar, T., Ward, N. D., Krusche, A. V., Richey, J. E., Yager, P. L., & Medeiros, P. M. (2016). Seasonal and spatial variability of dissolved organic matter composition in the lower Amazon River. Biogeochemistry, 131(3), 281-302.

Stubbins, A., Spencer, R. G., Chen, H., Hatcher, P. G., Mopper, K., Hernes, P. J., ... & Six, J. (2010). Illuminated darkness: Molecular signatures of Congo River dissolved organic matter and its photochemical alteration as revealed by ultrahigh precision mass spectrometry. Limnology and Oceanography, 55(4), 1467-1477

---

## Author Comment (AC2)

**Lyell Centre for Earth and Marine Sciences**

Dr James Spray
The Lyell Centre
Heriot-Watt University
Research Avenue South
Edinburgh
EH14 4AS, UK
Jfs8@hw.ac.uk
Tel: +44 131 451 3537

16th July, 2021

Dear Anonymous Reviewer 2,

Thank you for your review of our manuscript 'Unravelling light and Microbial Activity as Drivers of Organic Matter Transformations in Tropical Headwater Rivers', authored by Spray et al., which we submitted for publication in *Biogeosciences*. Please find below our responses to each of your comments in blue text.

Yours Faithfully,

Dr James Spray and Dr Ryan Pereira,

On behalf of all co-authors (Thomas Wagner, Juliane Bischoff, Sara Trojahn, Sevda Norouzi, Walter Hill, Julian Brasche, and Leroy James)

**Overview:**

In their manuscript, Spray and co-authors investigated the effect of light exposure and microbial activity on the degradation of dissolved organic matter (DOM) in tropical headwater streams. They conducted in-situ experiments, with containers fulfilled with unfiltered water and covered or not by a lid to mimic dark (microbial degradation only) or ambient (photochemical + microbial degradation) conditions. Containers were anchored at each site (3 sites), and experiments were performed three times. Changes in dissolved organic carbon (DOC) and DOM composition (using absorbance and size exclusion chromatography) were investigated after 12h of incubation. In order to separate the effects of photodegradation from microbial processing on DOC losses and changes in DOM composition, "the values from the 'ambient' container, which theoretically represent the combined effects of both photochemical and microbial influences, were corrected by subtracting the values from the 'dark' container" (lines 176-178). Based on this approach, the authors observed higher DOC losses in "ambient" treatments while no clear trend was identified upon microbial degradation processes. The authors concluded that photodegradation has a larger influence on DOC degradation than microbial activity.

The relative importance of photochemical mineralization versus microbial degradation is still an unresolved question, and thus the topic addressed by this study is relevant. The experimental setup is original in that sense that more classical studies use incubation in laboratory-controlled conditions, and therefore diverge from field conditions. However, I found several issues with the approach that may affect data interpretation and conclusions. I detailed my comments below and hope that they will be useful.

**General Comments:**

The structure of the manuscript could be improved significantly. In its current state, the results section contains a lot of interpretation. The authors could merge the results and discussion section as a Results and Discussion section, or better differentiate the two sections.

In response to these suggestions, and similar points from reviewer 1, we propose to better delineate the results and discussions sections as follows, with interpretations being moved to the latter section:

3. Results

3.1. Photochemical and microbial-driven changes in DOC concentrations

3.2. Photochemical and microbial driven changes in DOM composition

3.3. Organic Nitrogen Dynamics of DOM

3.4. Combined photochemical and microbial-driven changes over a day-night cycle

4. Discussion

4.1. Factors influencing photochemical and microbial-driven changes in DOC concentrations

4.2. Photo- and bio-lability of DOM component groups

4.3. Advantages and limitations of in-situ measurements

4.4. Implications and further research

My main concern is about the calculations made to separate photochemical degradation from microbial degradation outlined lines 176-183, where values from the 'ambient' container are corrected by subtracting the values from the 'dark' container. By doing so, the authors made the underlying assumptions that photochemical and microbial degradation are cumulative processes and that the microbial degradation of DOM is not affected by photochemical reactions. In other words, this approach suggests/implies that DOM microbial degradation is similar between "dark" and "ambient" treatments. However, there are clear evidence of interactions between photochemistry and biological degradation where sunlight exposure of DOM can either increase or decrease its bioavailability (Bertillon and Tranvik, 1998; Kaplan and Cory, 2016 for a short synthesis; Moran and Zepp, 1997). Given the experimental setup, the authors cannot exclude the fact that the dynamic of microbial degradation (both in terms of decay constants but also changes in DOM composition) differed across treatments. Therefore, I don't know what really mean the ambient – dark calculation from a conceptual point of view, but I am not convinced that it represents photochemical effects solely as interpreted by the authors along the manuscript (e.g. fig 2, lines 199, 206, 209-211, 225, 273, 275-277,...). In consequence, several interpretations and conclusions are not supported by the data presented. For instance, the authors concluded that there was no photo-stimulated biodegradation (lines 322-324) or that "photodegradation is a more influential process on the complete mineralization of DOC than biodegradation" (lines 418-420), yet the two processes cannot be so easily distinguish by the ambient-dark calculation. Maybe directly comparing treatments after incubations would be more appropriate to highlight the influence of light exposure on DOM degradation.

Our aim is to, as best as possible, disentangle net biotic from abiotic processes in close to 'real world' conditions, and in a remote setting where lab-based approaches are impractical. We are fully aware and accept that our approach, as any experimental and multi-proxy approach, has strengths and limitations. Our strength is the close to 'real' conditions, whereas one of our limitations may indeed lay in the not quantitative separation of photo-chemical and photo-stimulated microbiological turnover processes. We accept this fact and will work on the balanced wording of our text. The reviewer is correct to highlight that the photochemical effect we isolate may also incorporate immediate photo-stimulated microbial changes, but then other microbial control methods (mercuric chloride, autoclave, acidification and filtering) have been demonstrated to alter the DOM pool in some way and hence are equally not perfect – there is no one ideal approach. The practice of subtracting 'ambient' from 'dark' DOC loss in unfiltered river samples as a control may not be perfect and quantitative but has been applied before (Moody and Worrall, 2016) to successfully disentangle these effects. We note that in this previous study the authors' referred to this difference as "photo-induced degradation" rather than "photochemical degradation/photodegradation". We plan to pick up this revised definition and introduce the distinctions regarding DOC loss and compositional changes in our manuscript by modifying the last paragraph of the introduction:

"Approaches used to ascertain the importance of photodegradation and biodegradation of DOM in tropical headwaters isolate and incubate water, and establish how the DOC concentration and DOM composition changes over time, and by what dominant process (photochemistry versus microbial activity, or combinations thereof). These approaches tend to follow either laboratory-based methodologies, in which properties such as light and temperature can be controlled for, or are conducted in natural conditions, which are more representative but make disentanglement of multiple influences more difficult. In both approaches, photochemical and microbial influences are often separated by controlling for the latter by incubating a portion of the water sample in darkness. The further practice of treating and/or filtering the sunlight-exposed water prior to incubation can help to better control for rapidly occurring photo-induced biotic effects (e.g. Stubbins et al., 2010), but invariably alters the DOM pool, reducing representability. Incubating unfiltered, untreated water cannot control for these immediate photo-induced biotic effects, but can better characterise realistic net DOM changes (e.g. Moody and Worrall, 2016). Here the latter approach was adopted; unfiltered water samples collected from three sites in the headwaters of the Essequibo River, within the

Iwokrama Rainforest, Guyana (Fig. 1) were incubated in-situ in the river column to as close as possible achieve results representing actual in-stream processes. Splitting the initial sample and only exposing half to sunlight allows us to isolate and quantify net photo-induced processes from purely microbial ('dark') influences (Fig. 2a). Furthermore, by analysing the resulting samples using SEC, for the first time we can quantify not only changes in DOC concentrations, but also degradation-influenced compositional changes in DOM."

We also propose to stronger highlight the benefits and limitations of the approach in the newly proposed discussion (section 4.3), as follows:

"As outlined in the introduction, incubating unfiltered river water in natural conditions within the water column is beneficial in that it can quantify representative net changes in DOC concentrations and DOM compositions. It is also particularly appropriate for relatively remote locations, such as tropical headwater streams, where access to laboratories is limited. The decision to not treat water in the ambient containers to control for biota was taken to avoid altering the DOM pool, in an effort to maximise the representativeness of the results obtained and take advantage of SEC's ability to quantify a wide range of the DOM pool. The drawback of this approach, however is that the net photo-induced changes quantified likely include relatively rapid photo-stimulated microbial changes (Kaplan and Cory, 2016). In future, running a third incubation with poisoned or otherwise treated water could quantify this difference, while also revealing the effects of said treatment on the DOM pool as characterised by SEC."

Section 4.3 will also incorporate the discussions of POC-DOC, adsorption, sediment-hosted biotic processes, etc. detailed below.

Regarding our approach to quantifying 'photo-stimulated degradation,' again this was inspired by an approach and justification proposed by Moody and Worrall (2016). We recognise that this approach would be unable to quantify 'immediate' photo-stimulated biodegradation, but in the event that photoproducts were to accumulate in the water column under sunlight, then one could still expect to see some difference in the *overnight* behaviour in each container. It is still, in our opinion, therefore valid to test this concept, particularly regarding the lack of available data in tropical headwaters. Bertilisson and Tranvik (1998) found that in humic lake water, the production of photoproducts outstripped the rate at which they were consumed by bacteria during the day, leading to their accumulation in the water column (thereby leading to the possibility of the continuation of their consumption overnight). If such a trend were to have occurred at our sites, we believe that the approach taken would theoretically have been able to detect it. We propose to highlight this distinction in section 4.1 of the discussion (see new proposed structure for results and discussion) as follows:

"We explored the possibility that the accumulation of photoproducts during the day, as demonstrated by Bertilisson and Tranvik (1998) in humic lake waters, would lead to overnight photo-simulated biodegradation. Furthermore, it is possible that photoproducts may have been immediately taken up by biota in the 'ambient' container, as the water was not filtered or sterilized prior to incubation; rapid utilization may explain the scarcity of LMW acids during analysis (Fig. 3). A previous study with a similar methodological approach also found no evidence of overnight photo-stimulated bacterial degradation over a similar timescale in a temperate peatland headwater (Moody and Worrall, 2016)."

The interest of the experimental setup is that it is closed to field conditions. However, it is not possible to determine how other processes (POC-DOC exchanges, primary production) may affect the results obtained as water were unfiltered. Primary production (and DOM release) may also have been affected by light exposure. I would appreciate that the authors discuss the limitation of their study.

Our discussion highlights the point that, as stated by the reviewer, POC-DOC changes and primary productivity could have led to DOC concentration changes and some internal DOM fractionation. We

touch on this when framing the increases in DOC observed in the 'dark' incubations (line 217) and mention POC changes in the discussion (Lines 438-441). We propose to also expand this discussion to mention fractionation through adsorption, as follows:

"The approach presented here is limited in that it does not quantify POM-DOM interchange during the incubation process – filtering prior to incubation could circumvent this issue, but would reduce representability, whereas significantly larger samples would be needed to quantify POC. A further consideration is that adsorption to iron hydroxides within the water column could have fractionated the DOM pool to some extent; studies have shown that aromatics and polyphenols are more likely to adsorb and become stabilized, whereas aliphatic compounds preferentially remain in solution and are more easily destabilized once adsorbed (Coward et al., 2018)."

We also propose to discuss that this approach only quantifies 'in-water column' changes, with sediment-hosted microbial processes being absorbed:

"The findings presented here suggest that photodegradation has a higher potential for in-stream DOC removal than biodegradation, suggesting that the former may therefore be a more important driver of the degassing of $CO_2$ in tropical headwaters on daily timescales. However, further exploration is needed regarding microbial degradation processes hosted within sediment, which were not considered in the present study. "

These points, alongside an overall discussion of the limitations and benefits of the in-situ, unfiltered approach (see our response to first general comment above) will be consolidated into a dedicated section of the discussion (Section 4.3; see proposed restructuring of Results and Discussion).

Finally, a direct comparison of "photodegradation rates" with previous literature should be done with caution. It is hard to compare results obtained from different studies as the protocols may be very different, especially in this study.

We fully concur with the critical view of the reviewer. In our revised discussion of photodegradation rates (also see below) compared to other tropical rivers, we propose the following:

 "The mean degradation (5%) is comparable with those of a previous study on river water from the main stems of the Rio Negro, which reported DOC photodegradation of ~5% in unfiltered river water after ten hours of exposure to natural sunlight (Amon and Benner, 1996). It possibly exceeds that observed in the Congo River, which showed ~10% photodegradation of DOC over 48 hours of continuous exposure to a solar simulator (the intensity of which likely exceeds natural sunlight; Spencer et al., 2009); direct comparison is difficult however, given that river water in said study was filtered and poisoned prior to incubation."

When discussing the extension of our approach to other climate zones, we propose the following clarifications:

"The degree of DOC degradation observed in BC is higher than that observed in headwaters in other climate zones – like a boreal headwater (0.2 $km^2$ catchment) in Sweden (~10% decrease in DOC over 48 hrs (Köhler et al., 2002)) – but is lower than that observed in a temperate peatland headwater (0.2 $km^2$) in the UK (~40% decrease over 24 hrs (Moody and Worrall, 2016)) and two temperate forest streams in Japan (~21-36% over 12-13 hrs (Mostofa et al., 2007, 2005b)). These comparisons are hampered however by the lack of a consistent methodology used to quantify degradation in each case. Application of the technique demonstrated here to different climate zones and different river stages could therefore help to quantify how variations in DOC concentrations, environmental conditions, and DOM composition help to modulate degradation; whereas companion lab- and field-based incubation approaches conducted at the same sites could help to inform on differences in methodological approaches."

**Specific comments**

*Line 14:* "from "Line 15: the authors should present the mean instead of the higher values measured and used it to compare to previous studies (keeping in mind the limitation of such comparison). Mean DOC loss are 5%, and the 9% loss is more like an outlier than a real pattern.

*Line 201:* "The upper limit of this degradation exceeds those observed..." Please use the mean instead of the higher value, this is not representative.

We propose the following modification to the abstract: "…on average sunlight oxidised 5% of dissolved organic carbon (DOC) over 12 hours, at rates higher than or comparable to larger tropical rivers."

We have also propose the following edit to the results section (now in Discussion section 4.1) "The mean degradation (5%) is comparable with those of a previous study on river water from the main stems of the Rio Negro, which reported DOC photodegradation of ~5% in unfiltered river water after ten hours of exposure to natural sunlight (Amon and Benner, 1996). It possibly exceeds that observed in the Congo River, which showed ~10% photodegradation of DOC over 48 hours of continuous exposure to a solar simulator (the intensity of which likely exceeds natural sunlight; Spencer et al., 2009); direct comparison is difficult however, given that river water in said study was filtered and poisoned prior to incubation."

*Lines 130-137:* I would not have expected significant difference across treatments during the night since photochemical reactions and biological uptake are very rapid processes.

*Lines 210-211:* "non-aromatic compounds were preferentially removed by sunlight" ...or consumed by microbial communities. Overall, the ambient-dark calculation is interpreted as being solely due to photochemical processes along the manuscript, but this is misleading as biological activity was also occurring. This echoes my main concern about the approach used by the authors.

*Lines 322-325:* as said above, I am not surprised that no difference was observed during the night because photodegradation reactions are very quick, and labile compounds potentially produced are likely consumed also quickly and thus do not accumulate in the water column. Moreover, the authors cannot conclude that there was no photo-stimulated biodegradation due to the experimental setup and the limits of their calculation.

Please see our response to this issue in the general comments above.

*Lines 225-228:* Sites are located in the same river networks and close to each other, DOM collected at BBR comes from BC1 and BC2 headwaters as shown by similar initial composition (table 1 and figure 3), so it is not surprising that composition and reactivity are similar despite differences in drainage areas.

It is perhaps unsurprising that initial compositions are similar given the sites' closeness, though high temporal variability in DOC concentrations and organic matter composition have been observed in this region previously, as we highlight in the introduction (Pereira et al., 2014). We still believe that it is of note to highlight the different hydrological differences of each site, however, alongside our discussion of differences in cumulative irradiation and physical properties in the lines immediately following (Lines 229-242), with the proviso that this entire section will be split to better delineate the Results and Discussions sections as discussed above.

*Lines 456:* fig 6?

This should refer to figure 6 and we will update it accordingly (though note that this will become Fig. 5 as we are proposing to move the DON figure to the supplementary material in response to Reviewer 1).

*Figure 3:* there are errors in the caption.

We propose to correct the figure caption accordingly:

"**"Figure 3. a) Photochemistry-induced, and c) microbial-induced changes in the concentrations of photo-labile (HS, protein-based biopolymers and building blocks) and bio-labile compounds (non-protein-based biopolymers, LMW neutrals, and LMW acids) over 12 h; e) initial concentrations of photo-labile and bio-labile compound groups. B) Photochemistry-induced, and d) microbial-induced changes in the concentration of each DOM component over 12 h; e & f) DOM compositions of initial samples for each incubation. Note inverted y axes for (a-d)."**

References:

Amon, R. M. W. and Benner, R.: Photochemical and microbial consumption of dissolved organic carbon and dissolved oxygen in the Amazon River system, Geochim. Cosmochim. Acta, 60, 1783–1792, https://doi.org/10.1016/0016-7037(96)00055-5, 1996.

Bertilsson, S. and Tranvik, L. J.: Photochemically produced carboxylic acids as substrates for freshwater bacterioplankton, Limnol. Oceanogr., 43(5), 885–895,doi:10.4319/lo.1998.43.5.0885, 1998.

Kaplan, L. A. and Cory, R. M.: Dissolved organic matter in stream ecosystems: forms, functions, and fluxes of watershed Tea. in Stream Ecosystems in a Changing Environment, pp. 241–320, Academic Press., 2016.

Moran, M. A. and Zepp, R. G.: Role of photoreactions in the formation of biologically labile compounds from dissolved organic matter, Limnol. Oceanogr., 42(6), 1307–1316,doi:10.4319/lo.1997.42.6.1307, 1997.

Moody, C. S. and Worrall, F.: Sub-daily rates of degradation of fluvial carbon from a peat headwater stream, Aquat. Sci., 78, 419–431, https://doi.org/10.1007/s00027-015-0456-x, 2016.

Pereira, R., Bovolo, C. I., Spencer, R. G. M., Hernes, P. J., Tipping, E., Vieth-Hillebrand, A., Pedentchouk, N., Chappell, N. A., Parkin, G., and Wagner, T.: Mobilization of optically invisible dissolved organic matter in response to rainstorm events in a tropical forest headwater river, Geophys. Res. Lett., 41, 1202–1208, https://doi.org/10.1002/2013GL058658, 2014.